# Spectroscopic Analyses Highlight Plant Biostimulant Effects of Baker’s Yeast Vinasse and Selenium on Cabbage through Foliar Fertilization

**DOI:** 10.3390/plants12163016

**Published:** 2023-08-21

**Authors:** Ștefan-Ovidiu Dima, Diana Constantinescu-Aruxandei, Naomi Tritean, Marius Ghiurea, Luiza Capră, Cristian-Andi Nicolae, Victor Faraon, Constantin Neamțu, Florin Oancea

**Affiliations:** 1Polymers and Bioresources Departments, National Institute for Research & Development in Chemistry and Petrochemistry—ICECHIM, Splaiul Independenței nr. 202, Sector 6, 060021 Bucharest, Romania; ovidiu.dima@icechim.ro (Ș.-O.D.); naomi.tritean@icechim.ro (N.T.); marius.ghiurea@icechim.ro (M.G.); luiza.capra@icechim.ro (L.C.); cristian.nicolae@icechim.ro (C.-A.N.); victor.faraon@icechim.ro (V.F.); titi.neamtu@icechim.ro (C.N.); 2Faculty of Biology, University of Bucharest, Splaiul Independenței nr. 91-95, Sector 5, 050095 Bucharest, Romania; 3Faculty of Biotechnologies, University of Agronomic Sciences and Veterinary Medicine of Bucharest, Bd. Mărăști nr. 59, Sector 1, 011464 Bucharest, Romania

**Keywords:** plant cell wall, plant response, glycine betaine, molecular fingerprints, spectroscopic techniques, Fourier-transform infrared spectroscopy—FTIR, X-ray diffraction—XRD, thermo-gravimetric analyses—TGAs, soluble and insoluble fibers

## Abstract

The main aim of this study is to find relevant analytic fingerprints for plants’ structural characterization using spectroscopic techniques and thermogravimetric analyses (TGAs) as alternative methods, particularized on cabbage treated with selenium–baker’s yeast vinasse formulation (Se-VF) included in a foliar fertilizer formula. The hypothesis investigated is that Se-VF will induce significant structural changes compared with the control, analytically confirming the biofortification of selenium-enriched cabbage as a nutritive vegetable, and particularly the plant biostimulant effects of the applied Se-VF formulation on cabbage grown in the field. The TGA evidenced a structural transformation of the molecular building blocks in the treated cabbage leaves. The ash residues increased after treatment, suggesting increased mineral accumulation in leaves. X-ray diffraction (XRD) and Fourier-transform infrared spectroscopy (FTIR) evidenced a pectin–Iα-cellulose structure of cabbage that correlated with each other in terms of leaf crystallinity. FTIR analysis suggested the accumulation of unesterified pectin and possibly (seleno) glucosinolates and an increased network of hydrogen bonds. The treatment with Se-VF formulation induced a significant increase in the soluble fibers of the inner leaves, accompanied by a decrease in the insoluble fibers. The ratio of soluble/insoluble fibers correlated with the crystallinity determined by XRD and with the FTIR data. The employed analytic techniques can find practical applications as fast methods in studies of the effects of new agrotechnical practices, while in our particular case study, they revealed effects specific to plant biostimulants of the Se-VF formulation treatment: enhanced mineral utilization and improved quality traits.

## 1. Introduction

The plant cell wall is constantly monitored and remodeled by a complex network of cellular pathways—signaling cascades that respond to external and internal cues and regulate biodegrading and biosynthetic pathways [1,2]. Changes in the composition and architecture of plant cell walls occur as a plant cell response to various stresses [3,4,5,6]. These structural and compositional changes could represent an indicator of plant growing conditions, especially stress conditions and methods to alleviate them. To test this hypothesis, we applied a fertilizing product on cabbage, including a foliar fertilizer formula and plant biostimulants based on selenium salt and vinasse, as a tool to modulate plant response to stress. 

Plant biostimulants are a class of agricultural inputs developed in the 21st century and are defined by their agricultural functions/effects on cultivated plants, such as enhanced mineral nutrients uptake and utilization, increased cultivated plant tolerance to stress, and improved crop quality traits [7]. Plant biostimulants are considered an agronomic tool to mitigate abiotic stress [8,9,10]. 

Selenium (Se) is a plant-beneficial nutrient with a narrow physiological window [11,12,13,14], meaning that Se features beneficial effects in a dose range lower than one order of magnitude and dependent on the speciation (oxidation state and adsorbed forms) [15,16,17,18,19] being around 3–50 µM L^−1^ in certain formulations [20,21,22]. The application of selenium at beneficial doses under various forms (selenium salts, zerovalent nanoselenium, organic selenium) on plants, as soil or foliar treatments, determines effects similar to those defining plant biostimulants, the enhancement of mineral nutrients uptake [15,23], the utilization increased resistance to stress [24,25,26], and improved quality traits [27,28,29]. Selenium was mentioned in the category of inorganic plant biostimulants [7,30,31,32] and was widely recognized as a plant protection product against abiotic stress [24,33]. In the area with selenium deficit, selenium application combines biofortification, i.e., selenium supplementation of the food chains with the plant protection against abiotic stress—“protective biofortification” [34,35].

However, due to its narrow physiological window, selenium has also toxic effects on plants [36]. Low doses determine an antioxidant effect; by enhancing the activity of plant antioxidant systems, high doses increase the production of reactive oxygen species (ROS) and the toxic effect of ROS [37]. Selenium metabolism involves methylation and depletes the S-Adenosylmethionine metabolic pool [38,39]. The toxic effects are reduced by applying zerovalent nanoselenium particles [30,40], which are relatively non-toxic and act as a slow-release formulation due to a disproportionation reaction [16], i.e., redox reaction wherein one compound (e.g., Se^0^) acts both as reductant and oxidant and generates two reaction products, one with a higher oxidation state (e.g., Se^+4^) and one in a lower oxidation state—e.g., Se^−2^.

Our group developed a combination of selenium salts with glycine betaine as a methyl donor, which we tested on *Dunalliela salina* [41]. The added glycine betaine compensated negative effects of selenium on microalgae and further stimulated the accumulation of photosynthetic pigments and antioxidants in the *D. salina* biomass. For cultivated plant treatment, we used a more affordable source of betaine, baker’s yeast vinasse (BYV), and we integrated this composition with foliar fertilizers [21]. The application of this organo-mineral foliar fertilizer enriched with optimized selenium salt; vinasse doses demonstrated beneficial effects on the physiology of tomato plants and quality traits of the fruits, similar to those considered specific to plant biostimulants. As described in our previous study [21], vinasse is a general term for the by-product of yeast cultivation on molasses or raw sugar cane juices that contain up to 20% glycine betaine. Our baker’s yeast vinasse is the by-product of the baker’s yeast fabrication process. Glycine betaine is produced by plant metabolism as an osmoprotectant and has been shown to sustain plant response to different types of abiotic stress by stabilizing photosynthetic reactions and stimulating adenosine triphosphate (ATP) synthesis [42]. Exogenously applied glycine betaine acts as a plant biostimulant, increasing the plant tolerance to abiotic stress [43,44,45,46], enhancing nutrient uptake and nutrient utilization [47], and stimulating the accumulation of bioactive compounds in the edible yield [48,49]. 

We added the selenium salt and BYV composition to an NPK foliar fertilizer with microelement. Foliar fertilizers increase crop yield, especially in stressful conditions [50]. Nitrogen (N) is a constituent of cytoplasmic and membrane proteins, enzymes [51], reserve substances [52], nucleic acids [53], and chlorophyll [54]. Sufficient nitrogen supply accelerates plant growth and development [54,55,56], and nitrogen deficiency leads to the blockage of chlorophyll biosynthesis and leaf chlorosis [57,58,59,60]. Excess nitrogen stimulates photosynthesis, and the leaves become very large with low firmness, while plant root development is reduced [53]. 

Phosphorus (P) is also a major nutrient and is the main component of nucleic acids, phospholipids, and phosphoproteins [61]. It is involved in the chemical energy storage and transfer as a macroergic bond—e.g., in adenosine triphosphate, high-energy bonds used for storing and transferring free energy in biological systems. Phosphorus deficiency is characterized by excess production of anthocyanins [62], leading also to the diminishing of plant growth [63], while excess phosphorus may lead to leaf necrosis [64,65]. 

Potassium (K) plays an important role in osmoregulation [66] by changing its concentration in vacuoles, thereby controlling the cell turgescence, a phenomenon that underlies the process of closing and opening the stomata [67]. K increases plant resistance to frost [68,69] and drought [70], and it also stimulates the activity of enzymes involved in photosynthesis and respiration [71]. Potassium deficiency leads to chlorotic spots, which appear between the ribs or on the edge of the leaves [72], while excess potassium decreases root development and seedling growth [73]. The microelements B, Fe, Cu, Mn, Mg, Zn, and Mo were added to the foliar fertilizer to support further plant tolerance to stress and edible yield quality [74,75,76]. 

In the present study, we focused on the potential of the analytic fingerprints based on spectroscopic analyses (FTIR, XRD) to highlight the effects of the foliar fertilizing product that includes selenium–baker’s yeast vinasse formulation (Se-VF) on the cell wall structure from leaves of white cabbage *(Brassica oleracea* var. *capitata*
*f. alba*). We worked on dried leaf powder, which is a functional food additive and a dietary fiber source [77,78]. We combined spectroscopic analysis with thermogravimetric analysis and the determination of soluble/insoluble fibers to provide additional evidence for the modifications of plant cell walls induced by the treatment. These analytic techniques, less used in plant leaf characterization, demonstrate their capacity to reveal specific modifications of the leaf cell wall as indicators of plant growing conditions, and, particularly for our case study, the biostimulant effects on cabbage of the Se-VF formulation. These biostimulant effects revealed by the proposed spectroscopic techniques address both protection against abiotic stress and improve leaf dietary fiber composition. 

## 2. Results

### 2.1. Treated Plants Physiological Activity

The influence of the Se-VF on the growth and structure of white cabbage (*Brassica oleracea*, Capitata Group) was studied at two different doses, encoded D1 and D2 (D2 = 3 × D13 × D1), in comparison with the control experiment, C. The samples of the treated cabbage were denominated D1o and D2o for the outer leaves and D1i and D2i for the inner leaves. The physiological activity was monitored by measuring the chlorophyll fluorescence and stomatal conductance of the outer leaves. According to chlorophyll fluorescence measurements of the outer cabbage leaves presented in Figure 1a, after 2.5 weeks from the first treatment of cabbage with the Se-VF, there is a marginal improvement in the photochemical yield of PSII centers measured as chlorophyll fluorescence (0.6 ± 0.01% for D1o, and 0.58 ± 0.01% for D2o) compared to the control (0.52 ± 0.04% for Co). After 2.5 weeks from the second treatment, the lower D1 dose does not influence the PSII center efficiency, but there is an inhibition of PSII center efficiency for the D2 dose of foliar biostimulant (0.49 ± 0.03% for D2) in comparison with the first dose and control (0.59 ± 0.01% for D1 and C). 

The data in Figure 1b indicate a slight decrease in stomatal conductance 2.5 weeks after the second treatment for both tested doses (250 ± 65.6 mmol m^−2^ s^−1^ for D1, and 237.1 ± 40.8 mmol m^−2^ s^−1^ for D2) in comparison with the untreated batch (291.8 ± 34.7 mmol m^−2^ s^−1^ for C). 

### 2.2. Thermo-Gravimetrical Analyses–TGAs

Thermo-gravimetrical analyses of cabbage treated with Se-VF (dosage D1 and D2) and the control cabbage of the inner (Ci) and outer (Co) leaves were compared with standards of microcrystalline cellulose (MCC-Avicel), pectin (Pct), and alkali lignin (Lgn). Figure 2 and Table 1 evidence significant structural changes based on the thermal properties of biocompounds. TG analyses are discussed in terms of weight loss WL (%) vs. temperature (°C), and also as derivative weight loss vs. temperature. 

The temperature range of 25–525 °C was split into seven specific thermo-regions as a function of the main decomposing compounds: T_1_ = 25–105 °C was assigned to volatile organic biocompounds (VOBs) found in biomass samples together with free, interstitial, and weakly bound water (iwH_2_O); T_2_ = 105–160 °C was assigned to polyphenols, aminoacids/peptides, and strongly bound water (sH_2_O); T_3_ = 160–220 °C was assigned to proteins and some soluble fibers (xylose, glucomannans, gums); T_4_ = 220–260 °C was assigned to the rest of soluble fibers, pectins, hemicelluloses (which can also extend in the next region), and amorphous cellulose; T_5_ = 260–340 °C was assigned to crystalline cellulose; T_6_ = 340–420 °C was assigned to lignin as its main representative fraction; T_7_ = 420–525 °C was assigned to oxygenated biochar, containing aromatic carbonaceous structures rich in oxygen. Also, the residue was determined at 525 °C both in nitrogen and air after 30 min; this temperature was recommended for ash determination in the Megazyme Total Dietary Fiber assay procedure K-TDFR-100A/K-TDFR-200A 04/17.

In the first thermo-region, in T_1_ between 25 and 105 °C, the vaporization of superficial, free, interstitial, and weakly bound water, and volatile organic biocompounds (VOBs), like alcohols, aldehydes, terpenoids, and other plant volatiles, takes place [79,80,81,82]. The weight loss in this region varies from 3.87% for MCC, which is low-hydrophilic due to high crystallinity and also scarce in VOBs, to 4–5% for most of the vegetal samples. There is a 5.29% weight loss for lignin and a 7.33% WL for pectin, which is more hydrophilic due to polyphenolic, hydroxylic, and carboxylic functional groups. 

In addition to the WL of 3.87% in the first thermo-region, MCC has a high and representative WL of 90.40% in the thermo-region T_5_ (260–340 °C) assigned to crystalline cellulose, with an onset temperature maximum at 336.6 °C. Pectin is characterized by the thermo-region T_4_ 220–260 °C, the region assigned also to hemicelluloses and other fibers, with an onset maximum of 223.9 °C and a WL of 36.93%. Moreover, the commercial pectin has also a WL of 13.16% in the thermo-region T_5_ characteristic for cellulose, a WL of 5.46% in T_6_ characteristic for lignin, 4.12% in the biochar thermo-region T_7_, a 31.56% residue in N_2_ at 525 °C, and an ash content of 30.88% at the same temperature in air after 30’. Lignin, with its polyphenolic intercalated polymeric structure, has weight losses in all thermo-regions, which is further detailed and referred to, while its main representative thermal decomposition region is T_6_ (340–420 °C), where it has the maximum WL, 8.44%, at a T_max_ of 341.8 °C. Also, specific to lignin is the high residue at 525 °C, 61.96% in N_2_ and 61.91% in air, where the main core of aromatic rings is found thermally condensed in graphitic and other aromatic carbonaceous structures of biochar and ash. 

For the cabbage samples, inner and outer leaves and the controls are experimentally treated with the two dosages. The weight losses (WL) and the onset temperatures of specific thermo-regions offer suggestive information regarding the composition and structural features of cabbage leaves if compared in Figure 2 and Table 1 both vertically and horizontally. 

In the thermo-region T_1_, it can be seen that all samples, controls and fertilized, have higher WL in the outer leaves than the inner ones, which correlates with apparently higher carbohydrates (pectin, hemicellulose, and cellulose) and higher lignin content, suggested by the corresponding high residues (11–13% ash for inner leaves, 12–24% ash for outer leaves). These 4–6% weight losses up to 105 °C are related to low boiling-point biocompounds like alcohols, phenols, aldehydes, amines, ethers, and other volatiles, and interstitial and weakly bound water. The free water should be scarce in the samples, as they were analyzed in dry form. The temperature of vaporization on the other hand is lower for the outer compared with the inner leaves. The foliar treatment with Se-VF induces a significant effect in the T_1_ region on both the inner and outer leaves of cabbage with a clear increasing trend of percent and decrease in temperature (4.06%/74.2 °C, 4.57%/59.8 °C, 4.21%/59.7 °C for Ci, D1i, and D2i, respectively, and 4.37%/61.7 °C, 4.98%/50.7 °C, 5.29%/50.3 °C for Co, D1o, and D2o, respectively). 

In the next thermo-region, T_2_ 105–160 °C is assigned mainly to polyphenols and amino acids/peptides, but also sH_2_O can observe that the inner leaves have higher WL than the outer leaves. This region appears distinct on all thermograms, with a clear maximum of around 134 ± 5 °C for all cabbage samples. Se-VF induces an increase in WL in the inner leaves (6.26% for Ci, 6.35% for D1i, and 7.73% for D2i) and a decrease in WL in the outer leaves (6.08% for Co, 4.43% for D1o, 3.77% for D2o). 

In the third thermo-region, T_3_ 160–220 °C, assigned mainly to some soluble fibers and proteins, the inner leaves have higher WL than outer leaves, and Se-VF induces a decrease im WL in both the inner and outer leaves: 11.99% for Ci, 8.98% for D1i, 9.19% for D2i, 10.66% for Co, 7.99% for D1o, and 7.02% for D2o. The temperature has, in general, decreased in this region as a result of the treatment compared with the control. 

In the fourth region, T_4_ 220–260 °C, assigned mainly to soluble fibers, hemicelluloses, and pectins, the inner leaves have lower WL than the outer leaves. Se-VF determines an increase in WL in both the inner and outer leaves and a slight increase in the maximum temperature. 

In the fifth region, T_5_ 260–340 °C, which is characteristic of cellulose, the trend from the fourth region is maintained, i.e., an increasing trend of WL for all leaves, inner and outer, with an increase in the Se-VF dosage, and higher values of WL for outer leaves compared to inner leaves: 19.22% for Ci, 21.03% for Co, 20.34% for D1i, 21.26% for D1o, 21.29% for D2i, and 24.33% for D2o. 

The thermo-region T_6_ 340–420 °C was mainly assigned to the strongest structure, the condensed aromatic rings of lignin. The percentage is around 8% in all vegetal samples, with a slight increase in the outer leaves compared to the inner leaves. Part of this aromatic core is further found in the next region, T_7_, which is assigned to oxygenated biochar, which includes oxygenated hydrocarbons from cellulose and hemicellulose. 

The residue in nitrogen at 525 °C (T_8_) represents black carbon, meaning biochar was depleted in oxygen, especially oxygen from functional groups, which can still have calorific power due to oxygen sequestrated in heterocycles. The residues in N_2_ at 525 °C show a slight trend of higher WL values for the inner leaves compared with the outer leaves. 

After burning the residue in the air, the residue content has an opposite trend, with higher values in the outer than the inner leaves. The ash after burning in air (T_9_) shows mainly the mineral content of the samples, with a clear increasing trend toward fertilized samples and outer leaves compared with the controls and inner leaves. The samples are first to come in contact with Se-VF: 11.05% for Ci, 12.07% for Co, 11.10% for D1i, 15.14% for D1o, 12.92% for D2i, and 16.08% for D2o. 

### 2.3. X-Ray Diffraction Analyses—XRD

X-ray diffraction was applied, for the first time to the best of our knowledge, to characterize the cabbage leaves as native, unprocessed cabbage samples, and cabbage samples treated with a plant biostimulant. Related XRD studies mainly refer to cabbage-derived (nano)biomaterials, a few examples being the anthocyanins films from purple/red cabbage [83,84,85], activated carbon derived from cabbage leaves and wastes [86,87,88,89,90], and (nano)biocomposites using cabbage extracts or fibers [91,92]. In Figure 3a, the cabbage diffractograms were smoothed, translated, and overlaid with the diffractograms of microcrystalline cellulose (MCC, Avicel), pectin (Pct, Sigma-Aldrich), and lignin (Lgn, Sigma-Aldrich) for a first visual comparison of the main diffraction patterns. 

The cabbage diffractograms present a main peak around Bragg’s angle 2θ value 21.5° with a shoulder around 16.4° and a few small peaks later discussed. MCC has a main peak at 22.6° with a front shoulder around 20.7°, a second peak at 15° with a shoulder at 16.4°, and a third small peak at 34.6°, assigned to the convolution of cellulose Iβ peaks 14.83° (h,k,l) (−1,0,1), 16.42° (1,0,1), 20.21° (1,2,0), 22.71° (0,0,2), and 34.64° (−2,2,2) (PDF card No. 00-060-1502), together with small influences from cellulose Iα peaks at 10.28° (0,0,1), 14.26° (1,0,0), 16.77° (0,1,0), 21.80° (−1,1,0), 23.24° (0,1,1), and 25.04° (−1,1,1) (PDF card No. 00-056-1719) and amorphous cellulose peaks at 15.28°, 19.78°, 27.13°, and 36.10° (PDF card No. 00-060-1501). Commercial pectin has two main crystalline peaks, one at 13.2° with a sharp shoulder at 15.5°, and the second at 21.2°, while the amorphous peak is centered around 16.6° after deconvolution using Rigaku-PDXL 2.7.2.0 software. Commercial lignin has a broad amorphous peak centered around 21.5°, a quite strong peak at 34.7° with a front shoulder at 33.5° that might suggest some cellulose Iβ residues, and two small peaks at 28.1° and 41.8°. 

A first observation, best visible in Figure 3b,d with normalized intensities, is the main peak for cabbage samples (21.4°) for Ci—control inner leaves as representatives in Figure 3b are placed between the 21.2° peak of pectin and the 21.8° peak of cellulose Iα, suggesting that the biopolymeric structure of cabbage is based on pectin and cellulose Iα. 

In Figure 3c, the normalized diffractograms of cabbage samples are overlaid in order to analyze the differences between the treatments with the Se-VF foliar biostimulant. The diffraction spectra represent, as mentioned, a convolution of pectin, cellulose Iα, some cellulose Iβ, amorphous cellulose, and a small contribution from lignin, with a few particularities for certain samples. The first particularity is that the treated samples, D1i, D2i, and D2o, show a small new peak at 2θ = 8.9°. Another XRD particularity is that sample D1o has more shoulders and small peaks than the other samples; the shoulder at 16.3° is a possible convolution between amorphous cellulose (peak 15.3°) and cellulose Iα (peak 16.8°) and the shoulder at 26.6°, which is close to the amorphous cellulose peak 27.1°. Cellulose Iβ is present in all cabbage samples and is probably more concentrated in the ribs, a fact suggested by the contribution to the main peak around 21.5° and also the small peak around 34.8°. Another aspect is evidenced in Figure 3d by a close-up view of the main peak around 21.5°, placed, as mentioned, between the 21.2° peak of pectin and the 21.8° peak of cellulose Iα. The Co and D1o samples have the peak placed at 21.58°, closer to cellulose Iα, but Ci and all the other treated cabbage leaves, including D2o, showed a small shift of 0.12–0.16° toward pectin. 

The degree of crystallinity was also calculated for all samples as the ratio between the areas under the crystalline peaks and the total area, using Rigaku-PDXL 2.7.2.0 software. The crystallinity values of cabbage cultivars are placed between the lignin crystallinity of 34% and the pectin crystallinity of 60%, with values between 43% for Co and 58% for D2o, as can be seen in Figure 3a. A higher crystallinity is correlated with a higher content of pectin and cellulose Iα and a higher freshness and resistance of leaves, and a lower crystallinity is correlated with a higher lignin content. 

The crystallinity degree was higher in the control inner leaves (56% Xc) than the control outer leaves (43% Xc). The D1 dose has the opposite effect on the inner leaves than the outer leaves, decreasing by 9% and increasing Xc by 11%. The crystallinity now becomes 7% higher in the outer leaves (D1o) than the inner leaves (D1i). The 3× higher D2 dose increases with 4% more of the crystallinity of the outer leaves compared to the D1 dose and the decrease in Xc in the inner leaves is not as pronounced as the D1 dose, decreasing Xc with 5% compared to Ci. Interestingly, the difference between D2o and D2i is the same as the difference between D1o and D1i, 7%. Both doses result in the same difference in crystallinity between the inner and outer leaves. The increase in crystallinity suggests an increase in cellulose Iα and structural rearrangement of pectin in a more crystalline form. We shall discuss more about Xc when correlating with the FTIR data in the Discussion section. 

Figure 3d indicates that the inner leaves and D2o have a higher content of pectin than Co and D1o, which might suggest that the higher dose of Se-VF D2 induces pectin accumulation in the outer leaves (D2o). This is suggested by the slight shifting of the main cabbage diffraction peak from 2θ = 21.58° in the control Co to 21.42°–21.46° for the Se-VF-treated cultivars with the D2 dose, meaning toward the 21.2° pectin peak. The position of the deconvoluted amorphous peak in cabbage samples can also be considered relevant by comparison with the amorphous peaks of pectin (16.6°), cellulose (19.8°), and lignin (21.5°). The amorphous peak values presented in Figure 3a vary from 19.8°, 20.4°, and 20.5° in the inner leaves Ci, D1i, and D2i, respectively, to 20.6°, 20.9°, and again 20.9° in the outer leaves Co, D1o, and D2o, respectively. This suggests a higher content of amorphous cellulose in the inner leaves and a higher lignin content in the outer leaves. 

### 2.4. Soluble and Insoluble Fiber Contents 

We determined the soluble and insoluble fractions of cabbage leaves with the enzymatic kit available from Megazyme. For the insoluble fraction, the protein content was determined with the Kjeldahl method. The protein content was approx. 0.9% in the control and cabbage treated with the lower dose of Se-VF in the case of inner leaves and undetectable in the case of outer leaves. The protein content increased significantly to 1.3 and 1%, in inner and outer leaves, respectively, when applying a 3 L/ha dose. The percentages of dietary soluble and insoluble fibers are shown in Figure 4 and Table 2. Table 2 also contains the ratio between the soluble and insoluble fractions and the crystallinity determined from XRD. In the control samples, the insoluble fraction is similar in the inner and outer leaves, but the outer leaves contain a higher percentage of soluble fiber. The lower dose of treatment (1 L/ha) induces a significant increase in the soluble fraction accompanied by an increase in the insoluble fraction in the inner leaves, but the outer leaves are not influenced significantly. By increasing the dose to 3 L/ha, the soluble fiber increases further, and the insoluble fiber starts to increase in the inner leaves compared to the 1 L/ha dose. The higher dose has a slight and reversed effect on the outer leaves compared to the inner leaves, i.e., the soluble fraction decreases, and the insoluble fraction increases slightly but not significantly. 

The ratio between the soluble and insoluble fraction increases significantly as a result of treatment in the inner leaves, and it is less affected by the treatment in the outer leaves, where a slight decrease is observed at the higher dose applied. The trend of the ratio values correlates with the crystallinity determined by XRD. 

### 2.5. Elemental Composition and Total Protein Content

There are no significant differences in the C content of the inner leaves (37 ± 0.1% in Ci, 37 ± 0.5% in D1i, and 36.5 ± 0.2% in D2i). In the outer leaves, the C content is lower than the inner leaves, and there are significant changes after the applied treatments. Untreated outer leaves (Co) have a C content of 29 ± 0.1%. There is a significant increase in C content after the D1 treatment up to 34.8 ± 0.07% (D1o), followed by a significant decrease to 30.7 ± 0.2% (D2o) after the D2 treatment. 

In terms of hydrogen content, there were no significant differences between the D1 and D2 treatments in the inner leaves (5.37 ± 0.04% in D1i, and 5.44 ± 0.02% in D2i), but a marginally significant decrease was observed compared to the untreated control (5.52 ± 0.1% in Ci). The untreated outer leaf control Co has lower H content than the inner leaf control (4.46 ± 0.02% in Co). There is a significant increase in H content after D1 treatment (5.2 ± 0.002% in D1o, followed by a significant decrease after D2 treatment reaching the control level (4.5 ± 0.02% in D2o). 

Between D1i and Ci, there are no significant differences in the total nitrogen content (4 ± 0.1% in Ci, and 4.08 ± 0.1% in D1i), but in D2i, the N content increases significantly up to 4.6 ± 0.06%. Co has lower nitrogen content than Ci, i.e., 3.2 ± 0.1%, and there is a significant increase in N content following D1 and D2 treatments (4.40 ± 0.02% in D1o, and 4.30 ± 0.10% in D2o) (Figure 5a). The C/N ratio decreased following the treatment compared with the control, especially at the D2 dose (Figure 5b). This implies that the N content increases more than the C content. The data correlate with the increase in protein content following the D2 treatment, but other compounds rich in N probably contribute, including nitrates. 

### 2.6. ATR-FTIR Spectroscopy of Cabbage Samples

Attenuated total reflectance Fourier-transform infrared spectroscopy (ATR-FTIR) is an analytical method that could be used to convert the light absorbed by the structural molecules of plants into a characteristic absorption spectrum that can be interpreted as a biochemical fingerprint. ATR-FTIR spectroscopy was employed in the present study to evaluate the IR spectral signals of cabbage leaves. MCC, Pct, and Lgn were also analyzed using ATR-FTIR in order to identify characteristic absorption bands of the main structural macromolecules found in cabbage. The control spectra of MCC, Pct, and Lgn were recorded and overlaid in Figure 6, where the main characteristic bonds are shown. The figure is also shown in Appendix A as Appendix A, together with a detailed characterization of the spectra bands. 

FTIR spectra can reveal the characteristic bond vibration of functional groups at specific frequencies/wavenumbers, representing an analytical method considered fast and relevant since its early days, especially for enzymatic reactions [93]. It is also useful in biochemical fingerprinting and the taxonomical discrimination of plant species [94]. The mid-IR spectrum, also called the thermal infrared energy range [95] and defined by the wavelength interval of λ = 2.5–25 µm, corresponding to wavenumbers ν = 10,000/λ = 4000–400 cm^−1^, is usually divided into two main regions: the diagnostic region within ν ≈ 4000–1500 cm^−1^, characterized by higher IR energies, and the fingerprint region within the range of ν ≈ 1500–400 cm^−1^ at lower IR energies, inducing characteristic vibration-rotation movements of chemical bonds and functional groups. The FTIR spectra of the cabbage samples Co, Ci, D1o, D2o, D1i, and D2i were compared in Figure 7. Co was also compared with MCC—microcrystalline cellulose, Pct—pectin, and Lgn—lignin in Appendix A in order to identify similarities with the structural relevant macromolecules. The comparison between cabbage and MCC, Pct, and Lgn in Figure 6 and Figure 7 and Appendix A evidence that the IR spectrum of cabbage is a convolution of all three major macromolecules with additional particularities. Pectin has a significant contribution to the cabbage spectrum, exemplified by the bands at 3853(Ci)/3838(Co) cm^−1^ and 3740 cm^−1^ for the free hydroxyl groups, the band at 1738 cm^−1^ for the C=O vibration (esterified and protonated pectin), 1616(Ci)/1603(Co) cm^−1^ for carboxylate groups, and 1221 cm^−1^ for esters. Cellulose influence is observed in the region of bound hydroxyls with a peak at 3286(Ci)/3281(Co) cm^−1^ and also in the polysaccharides band with a peak at 1028 cm^−1^. The lignin influence appears mainly as shoulders with low intensity, suggesting smaller contributions and the role of a binding substrate for cellulose, pectin, hemicelluloses, proteins, and other biomolecules, the absorption band at 3024 cm^−1^ for aromatic C-H vibrations, the shoulder at 1556 cm^−1^ for the contribution of C=C vibration in monolignols, a contribution in the polysaccharides region with aromatic C-H in-plane deformation (1138 cm^−1^), and a band for aromatic ring breathing at 521(Ci)/523(Co) cm^−1^.

Pectin, (hemi)cellulose, and lignin were previously reported to be constituents of cabbage leaves; the pectin/hemicellulose content was the highest and the lignin content was much lower among the three constituents [96,97,98,99,100,101]. We also performed the extraction of pectin, cellulose, and lignin from untreated cabbage and validated the extracted components by comparing the FTIR and XRD analyses with commercial samples (Appendix A). In general, the extracted biopolymers had similar features to the commercial ones with some particular differences, the most noticeable being the much lower crystallinity and the absence of the second peak at 13.2° in the extracted compared to commercial pectin, confirming the XRD pattern of the cabbage leaves (Figure 3). 

Other strong spectral signals that cannot be explained by MCC, Pct, or Lgn contribution are the strong, sharp peaks for C-H at 2918 and 2852 cm^−1^ in the diagnostic region and the bands for C-C-H at 1367 and 457 cm^−1^ in the fingerprint region, which suggest the presence of long aliphatic chains that are probably from aliphatic S-glucosinolates (β-thioglucoside-N-hydroxysulphates) [80,102,103], possibly including Se-glucosinolates [104,105,106,107,108,109], lipids [110,111,112], and/or hydroxyketones [113,114]. 

The influence of Se-VF was investigated based on the spectra in Figure 7 for inner and outer leaves treated with D1 and D2 and was compared with the control C. The first observation can be made for the 3900–3600 cm^−1^ domain, where the absorption bands of free OH groups appear. It is observed that the representative absorption peaks of 3740 cm^−1^ and 3848 cm^−1^ are much more intense for the control sample (Ci) and decrease relatively to the other peaks by increasing the Se-VF concentration, a possible hypothesis that is correlated with a lower number of free OH functional groups and a higher number of bound OH groups in D1i and D2i compared to Ci. The peak intensities in the absorption area of 3650–3000 cm^−1^, characteristic for bound OH groups and N-H bonds are significantly higher in the D2i sample compared to Ci and D1i. 

The next diagnostic region, 3000–2800 cm^−1^, is characteristic of C–H bonds of aliphatic chains, like in lipids, glucosinolates, and hydroxyketones [102,112,115,116,117], but also some carbohydrates with strong, sharp bands, like the ones in Figure 7a at 2918 and 2852 cm^−1.^

The next significant absorption band in the cabbage samples is the band around 1738 cm^−1^, which is specific for the stretching vibration of the C=O bond, which, as evidenced in Figure 6, Figure 7, and Appendix A and Appendix A, is characteristic of pectin carboxyl and methylester groups/esterified uronic acids of hemicelluloses. A more intense peak around 1735 cm^−1^ in *Brassica napus* suggested the accumulation of pectin in the cell wall [115] and more exact esterified and/or protonated pectin [118,119,120], which, in our case, in Figure 7, points to the controls Ci, especially Co. The maximum peak around 1610 cm^−1^ is assigned to the ionized form COO^−^ [119], which overlaps with amide I (1650 cm^−1^) and amide II (1550 cm^−1^). The relative proportion of the two bands was evaluated in Appendix A by peak integration of the band area around 1740 cm^−1^ and the band around 1610 cm^−1^ [118,119,121,122,123,124]. 

The degree of methylesterification (DM, %) or esterification (DE, %) in pectins was proved to be assessable by integration of the FTIR bands around 1740–1750 cm^−1^ for esterified or protonated pectin and the band around 1610–1630 cm^−1^ for carboxylate, unesterified pectins, which were sometimes in good agreement with other methods like titration, HPLC, and GC-MS [118,119,121,122,123,124]. The degree of esterification is, in general, proportional to the ratio between the two bands. Indeed, the evaluation by FTIR of the ester, uronate, and amide content in pectin and alginate was proposed in 1975 by the deuteration of various samples and correlated with the titration method, HPLC, and atomic absorption spectroscopy [123]. Nevertheless, some studies have shown that due to the overlap of esterified and protonated forms, the degree of methylation can sometimes be overestimated [119]. The integration of the peak at 1738 cm^−1^ represents, in this case, the proportion of esterified (methylated) and protonated polysaccharides (which we denote here as DMP). Nevertheless, we believe that the protonated form in our case is not in a significant proportion because this would have led to a more significant peak around 830–880 cm^−1^, as previously reported for COOH [120]. Although the amide I band around 1650 cm^−1^ overlays with the carboxyl band around 1610 cm^−1^, in Appendix A, the second integration area includes both bands for all samples, and the calculated DMP can be still considered suggestive to characterize the cabbage samples, as the protein content was determined to be around 1% (*w*/*w*). Moreover, for the commercial citrus pectin (Pct, galacturonic acid ≥ 74.0%, Merck-Sigma Aldrich), the DMP value of 45.09% was close to the 47%, 47.6%, and 48% DM values obtained by titration, FTIR integration, and HPLC for the same commercial compound [121]. The DMP values presented in Appendix A suggest that the foliar treatments with Se-VF induce a stabilization of pectins in unesterified/deprotonated, ionic forms. 

The acidification of samples with 1M H_2_SO_4_ and the obtained FTIR spectra presented in Appendix A showed the shift of the peak maximum from 1610–1616 to 1638 cm^−1^. The 1738 cm^−1^ peak shifts to lower wavenumbers, around 1720 cm^−1^, probably reflecting the higher contribution of protonated carboxyl. The two peaks (1720 and 1638 cm^−1^) are almost of the same intensity in most of the acidified samples. The acidification protonates the ionic form COO-, which leads to a significant reduction in the corresponding peak at 1610–1616 cm^−1^. The amide band becomes more intense and equals or even surpasses the 1720 cm^−1^ peak. The data show that the equilibrium shifted toward the carboxylic form, a fact also confirmed by the significantly more pronounced bands at 873 cm^−1^ compared with the non-acidified samples. These data suggest that before acidification, the maximum at 1738 cm^−1^ was dominated by esterified carboxyl. 

The 1750–1200 cm^−1^ absorption domain is specific for amides and amines bands [115,116,125,126], such as amide I in the 1720–1600 cm^−1^ range for the C=O bonds with peaks around 1630 cm^−1^ [115,125], amine I and amide II in the 1600–1500 cm^−1^ region for the C-N bonds with peaks around 1560 cm^−1^ [115,116,125,127], and amine II and amide III bands for the C-N-H vibrations in the 1350–1200 cm^−1^ region with peaks around 1240 cm^−1^ [125,128,129]. Amine I and amide II bands overlap with the deformation vibrations of the –OH groups in cellulose (1645–1600 cm^−1^) [94,126,130] and with the C=C bond vibration range in aliphatic or aromatic structures (1660–1580 cm^−1^) [116,126,128]. In Figure 7, it is observed that for the control sample, the main peak at 1616(Ci)/1603(Co) cm^−1^ has two shoulders at 1574 and 1522 cm^−1^, while the samples treated with Se-VF convolute in an intense peak, especially in the case of treatment with D2. This fact might suggest an increased involvement in the amidic and carboxyl groups in intermolecular bonds together with possible chelation with cationic nutrients; the –COO^−^ salts have a specific absorption band in the region of 1610–1550 cm^−1^ [126,128,131]. 

The spectral area of 1485–1185 cm^−1^ corresponds to structural carbohydrates (STCHO) [132] and is mainly represented by cellulose, hemicellulose, pectin, and lignin, and shown in Figure 7 are the three main peaks around 1423 cm^−1^ (δC-H asymmetric), 1367 cm^−1^ (δC-H symmetric), and 1221 cm^−1^ (νC-OH), similar to ca. 1415, 1374, and 1234 cm^−1^ in *Brassica carinata* and *Brassica napus* [132]. The 1 L/ha dosage of Se-VF does not produce major effects on the molecular structure of the treated cabbage reflected in this region. The 3 L/ha dosage produces a significant decrease and broadening of the absorption band at 1221 cm^−1^.

The next band, 1200–850 cm^−1^, is known as the polysaccharide or carbohydrate band [132] and consists of different C-O, C-O-C, and C-OH vibrations. In this region, sample D2i has the most intense band with a peak at 1034 cm^−1^, a front shoulder around 1142 cm^−1^, and a tail shoulder around 874 cm^−1^ assigned to C-O, C-O-C, and C-OH in structural carbohydrates. The region, 1035–1041 cm^−1^, has also some contribution from lignin, as can be seen in Figure 6 and reported previously [133], but in our case, the lignin content seems to be less influenced compared to the other peaks that are characteristic of lignin, which are not significantly affected. We used the maximum peak in this region, in our case 1034/1018 cm^−1^, to obtain a crude assessment of the esterification of cell walls by dividing the area of the 1738 cm^−1^ peak by this area [134]. The same trend as for the 1738/1612 cm^−1^ ratio was obtained. 

Conclusively, from FTIR spectra, it appears that Se-VF mainly stabilizes pectin in low-esterified form, increases the lipids and/or aliphatic glucosinolates and carbohydrates in cabbage leaves, and biofortifies the molecular structure by increasing the intermolecular H-bonding. 

## 3. Discussion

Foliar fertilization is an increasingly applied method aiming to deliver nutritive solutes through the leaves of plants, for plant protection and nutrition [135,136,137,138,139,140]. Thereby, an organo-mineral selenium-enriched glycine betaine extract from baker’s yeast vinasse (BYV) was formulated together with NPK+ minerals as a complex foliar fertilizer after a similar protocol previously reported with a lower GB content tested on tomato cultivars [21]. For tomato plants, our previous Se-BYV fertilizer showed a growth promotion effect, especially for the 6–8 µM Se concentrations, a modulatory effect of nutrient uptake, and an increased stomatal conductance and photosynthesis efficiency. Regarding the tomato fruits, the initial Se-BYV fertilizer produced an increased number of top-quality fruits and also induced a significant accumulation of bioactive compounds [21]. 

Both doses, D1 and D2, used in this study have similar effects on the photosynthetic activity after the first treatment, slightly improving the activity compared to the control. Most of the amount of water lost by the plant through respiration and evapotranspiration occurs through the stomatal apparatus [141]. Water is expelled by the mesophyll into its intercellular spaces, and then water vapors diffuse from the mesophyll through the ostioles into the atmosphere. So, the stomatal aperture is the dominant driving factor of diffusion at the leaf surface, which controls both CO_2_ absorption and plant leaf water loss in the photosynthesis process [142]. By measuring the degree of stomatal aperture as a function of the velocity of a gas passing through a leaf segment, our data (in Figure 1b indicate a decreasing tendency in stomatal conductance 2.5 weeks after the second treatment for both tested doses (250 ± 65.6 mmol m^−2^ s^−1^ for D1, and 237.1 ± 40.8 mmol m^−2^ s^−1^ for D2) in comparison with the control (291.8 ± 34.7 mmol m^−2^ s^−1^ for C). Under the influence of our Se-VF, stomatal conductance slightly decreases in a dose-related manner, suggesting a tendency to slightly increase water use efficiency [35,143]. 

The thermo-gravimetrical analyses suggest an induced remodeling of the cabbage cell walls by the applied treatments, as presented in the following. The content of the main components is influenced by the treatment. The weight loss variations in the nine proposed thermo-regions suggest an accelerated development of cell walls by foliar treatments with Se-VF. The TGA is also confirmed by the FTIR and XRD results on pectin and cellulose Iα accumulation in the particular T_4_ and T_5_ thermo-regions presented in Table 1.

The TGA data indicated higher content of VOC and/or interstitial/weakly bound water in the outer than the inner leaves, as evidenced in the first thermo-region, which correlated with a decrease in thermal stability through the promoting of the ignitability mechanism [144,145]. This difference between the inner and outer leaves could be related to the different functions. The outer leaves are more exposed to the environment and have a protective role; therefore, there is a need to synthesize more VOC. Se-VF induced an increase in VOC and a decrease in T1 in both the inner and outer leaves, which suggested the stimulation of biosynthesis of small biomolecules with lower boiling points. This enhances the resistance of plants to abiotic and biotic stress and proves the biostimulant character of the treatment. Lower T1 could be also caused by more loosely bound water. 

The second thermo-region of the TGA, T_2_, between 105 and 160 °C, was assigned mainly to polyphenols, amino acids, (oligo)peptides, and other complex products photo-synthesized by plants from the initial molecular building blocks, and also strongly bound water. This region suggests a more intense plant metabolism in the inner than in the outer leaves, with higher content of polyphenols and amino acids. The higher content of strongly bound water in the inner compared to the outer leaves could also explain the difference. Se-VF induces an even higher content in the inner leaves and a lower content in the outer leaves compared with the control, and the difference between the two types of leaves increases significantly. This shows that the treatment fortifies the inner leaves against the outer leaves with certain organic compounds.

The third thermo-region, T_3_, 160–220 °C, was assigned to proteins and soluble fibers, which are also called dietary fibers, referring to vegetal fibers that are soluble in water and gastro-intestinal fluids, like β-glucans, galactomannans, soluble pectins, lipids, and gums like guaran, oligosaccharides, and plant hormones. All inner leaves have higher WL than outer leaves, which suggests that the biochemical processes that synthesize soluble fibers and/or proteins are more intense in the young leaves. This can also be seen as an intermediary biosynthesis step in the development of the plant, i.e., soluble fiber replacement with insoluble, and structural fibers, like cellulose and lignin. This hypothesis is confirmed by the next two thermo-regions, hemicelluloses (T_4_) and celluloses (T_5_), which both show higher WL values in the outer leaves than the inner leaves. With this hypothesis in mind, we can look again at thermo-region T_3_, where the cabbage control has higher WL values (11.99% for Ci and 10.66% for Co) than the fertilized samples (8.98% for D1i, 7.99% for D1o, 9.19% for D2i, 7.02% for D2o) and discuss the influence of our Se-VF. As proposed, if we see the prevalence of soluble fibers as an intermediary step in plant development (from young to old), it means that our Se-BNF accelerates the plant metabolism through the higher biosynthesis of volatile metabolites, polyphenols, amino acids, and peptides (T_1_-T_2_).

As mentioned, the T_4_ region, 220–260 °C, was assigned to the thermal decomposition of hemicelluloses, pectins, and amorphous cellulosic short chains. In this region, MCC has no WL, commercial lignin has a WL of 6.34% due to its polyphenolic structure, and commercial pectin has the highest WL of 36.93%, which is, therefore, considered as its characteristic region. For the cabbage samples, the trend of higher WL is clearly visible, as well as higher pectin and hemicellulose content in outer leaves with respect to the inner leaves, and also higher content in the foliar-fertilized samples compared to the control. This suggests that the biosynthesis of the apoplast and primary cell wall is still young in the inner leaves and more developed in the outer leaves but accelerated by Se-VF in both types of leaves (8.38% in Ci, 9.68% in Co, 11.06% in D1i, 12.85% in D1o, 9.59% in D2i, 12.55% in D2o). We notice that the values for the second dosage, D2, are smaller than the ones for D1 Se-VF. This observation, correlated with the next thermo-region, T_5_, where D2 presents the highest cellulose content (21.29% for D2i and 24.33% for D2o, compared to 20.34% for D1i, 21.26% for D2o, 19.22% for Ci, 21.03% for Co), supports the proposed hypothesis of intensified metabolism induced by Se-VF. This means that the cabbage cultivars treated with the highest dosage of Se-VF stimulated a faster plant metabolism toward the secondary cell wall, which was built mainly from cellulose chains. The onset temperatures in the T_4_ region are around 238 ± 2 °C, with a slightly increasing trend in the outer leaves, suggesting higher lignin content, but it is still significantly lower than the onset temperature of lignin in this region, 259.1 °C, and is closer to pectin onset temperature 223.9 °C, suggesting high amount of pectin together with hemicellulose inside the matrix of middle lamella of the cell wall, especially in young leaves. 

The thermo-region T_5_ is characteristic to mature, structural cellulose and secondary cell walls in plants. As we mentioned, the effect of Se-VF on the structural carbohydrates-skeleton of studied cultivars is clearly visible in this region. The previously mentioned accelerated metabolism is clearly visible in this region, with the outer and treated leaves having higher WL than the inner and non-treated leaves. 

The Se-VF mineral nutrition of cabbage is confirmed by the increased ash content in the inner leaves at around 17%. It is around 33% in the outer leaves following the D2 treatment. Part of the ash increase is attributed to the lignin content, which is higher in the outer leaves, as evidenced by XRD. 

The XRD data show that the biopolymeric structure of cabbage is based on pectin and cellulose Iα, not cellulose Iβ, as was probably expected for higher plants. This aspect can be correlated to some extent with the edibility aspect, the Iα allomorph being metastable [146]; therefore, it is more easily degradable than cellulose Iβ. 

The one-chain triclinic system of cellulose Iα is known to be characteristic of bacterial cellulose, tunicates, some algae, and cellulose Iβ, with a two-chain monoclinic system that ensures a more rigid structure for higher plants [130,146,147]. 

The prevalence of cellulose Iα in cabbage can be considered the first relevant XRD fingerprint, and apparently, the first reported substantiated evidence of cellulose Iα dominance in cabbage leaves. The XRD patterns of cabbage were not of particular interest until now, in our state-of-the-art survey, while the presence of cellulose Iα in Chinese cabbage fibers was previously suggested in a discussion on FTIR spectra [92], but without particular Iα band assignments and referring only to a study on cellulose crystallinity that did not mention cellulose Iα [148]. Additionally, in the mentioned study [92], the peak at 2θ = 22.8° was used for X-ray crystallinity, which is generally assigned to cellulose Iβ in Avicel, MCC, cotton, or wood [149,150,151,152]. 

The peak appearing at 8.9° following treatment with Se-VF is intriguing. This angle is specific for the 001 diffraction plane. The XRD database gives methylcellulose as a possible compound at 8.52° (PDF card No. 00-062-1291), but it is not known to exist as a natural compound in plants. The angle also appears in some clay minerals such as phlogopite, biotite, and muscovite, but with sharper peaks than ours, which is characteristic of crystalline structures [153]. It is also characteristic of graphene oxide, and diffractions in this region have been obtained for some lignin nanoparticles and other carbon-based nanoparticles [154]. It is possible that the appearance of this peak reflects some structural changes in the biopolymeric organization of cabbage leaves, but further evidence is needed for confirmation. 

Considering the molecular complexity of foliar composition and the uptake of nutrients by plants, we aimed to find relevant analytic fingerprints in FTIR spectra for assessing the composition and structural changes in biofortified cabbage obtained by treatment with Se-VF. ATR-FTIR is a fast, simple, reproducible, non-invasive, and cost-effective analytical method by which plant structural molecules like polysaccharides, proteins, lipids, and nucleic acids, together with secondary metabolites like phenols, terpenoids, alkaloids, and the intra- and inter-molecular interactions in the plant cell wall can be studied [94,155,156]. FTIR spectroscopy is a renowned molecular fingerprinting method, which has been previously used to show that cabbage and other related vegetables have IR spectra characteristic to a convolution of cellulose, pectin, and lignin, with additional particularities represented by aliphatic biocompounds like glucoiberin, gluconapin, glucoraphanin or sinigrin glucosinolates [80,102,103], lipids, phospholipids, glucolipids [110,111,112,157], and possible hydroxyketones [113,114]. FTIR proved very useful in analyzing the influence of fertilizers on other plants from *Brassica* genus. For example, *Brassica oleracea* L. var. *botrytis* treated with biochar and 130 and 260 kg N/ha showed stronger absorption bands for phenols and proteins in the H-bond region around 3290 cm^−1^ for carboxylic acids, lipids, and carbohydrates around 2860 cm^−1^, and for sulforaphane from glucoraphanin around 1400 and 1051 cm^−1^ [102]. 

The absorption bands of free OH are generally assigned around ν = 3750–3600 cm^−1^ [128] or around λ = 2.70 µm (ν = 3704 cm^−1^) [158]. The stretching vibrations of bound -O-H, associated via hydrogen bonds, appear in the broad region of 3600–3000 cm^−1^ [93,128,159,160,161,162]. An interesting aspect regarding the hydrogen bonds of cellulose in this region was evidenced in 1954 by Marrinan and Mann [160] using the deuteration with D_2_O of various cellulose samples, revealing that amorphous cellulose is more reactive than crystalline chains and gets involved in hydrogen bonds with D_2_O, displacing a part of the -OH absorption band from 3450 to 2550 cm^−1^, an observation strengthened in 1957 by Tsuboi [161]. Moreover, the remaining four absorption bands of crystalline cellulose in the region of 3600–3000 cm^−1^ were assigned by Marrinan and Mann [160] to single hydrogen bonds O-H…O around 3484 cm^−1^ and 3444 cm^−1^, respectively, to double hydrogen bonds H…OH…O around 3322 cm^−1^ and 3163 cm^−1^. In this region, MCC (Avicel), which is a type of partially depolymerized cellulose I, has a peak maximum at 3331 cm^−1^ and is assigned to the vibration of -OH involved in H-bonds, and one shoulder at 3279 cm^−1^ is assigned to a particular Iβ allomorph, while there is an additional shoulder for Iα appears around 3240 cm^−1^ for Iα-rich celluloses, like bacterial cellulose and some algae [163,164]. Regarding cellulose IA and IB, now better known as Iα and Iβ, it was then reported that the absorption band at 3242 cm^−1^ (IA) is present in the bacterial cellulose produced by *Acetobacter acetigenum* and *A. xylinum*, and also in alga *Valonia ventricosa*, but it is undefined in plant celluloses spectra, while cellulose IB has the band visible around 3275 cm^−1^ [163]. Similar peak values were later reported; the bands at 3270 cm^−1^ and 710 cm^−1^ were assigned to Iβ (@3279 and @706 cm^−1^ in Figure 6 for MCC) at 3240 cm^−1^ and 750 cm^−1^ for Iα [146,164] (@3238 cm^−1^ and @749 cm^−1^ in our previous study on bacterial cellulose [130], although they were not particularly assigned as Iα. The Iα + Iβ semi-crystalline composite structure of native celluloses was confirmed in 1984 using the solid-state ^13^C nuclear magnetic resonance and enriched with the spectral particularities of all six anhydroglucose carbons [165,166]; and in 1991, it was established by electron diffraction that the Iβ unit cell is composed of two chains in monoclinic arrangement, while an Iα unit cell is one-chain triclinic [167]. 

For our Se-VF-treated cabbage, in the first FTIR fingerprint, the decreased bands of free -OH groups around 3740 cm^−1^ and 3848 cm^−1^ correlated with the increased band around 3281 cm^−1^ of bound -OH, especially for the D2 dosage of 3 L/ha (10.5 µM Se), suggesting a biofortified molecular structure with higher proportion of H-bonds, especially the intermolecular H-bonds of carbohydrates [115]. For the D1i sample, which was treated with a 3× lower dosage than D2, this effect is lower. 

A second relevant FTIR fingerprint was considered in the intense C-H bands around 2918 and 2852 cm^−1^ for treated cultivars, which may indicate more aliphatic glucosinolates and/or lipids [102]; the presence of Se-glucosinolates was also possible [104,105,106,107,108,109], but was not confirmed in the present study. By analyzing the FTIR spectrum of BYV presented in Appendix A, we observed that the C-H band around 2926 cm^−1^ is not as sharp as the ones in Figure 7 for D2i and D2o, so there is not a physical deposition effect of BYV on cabbage outer leaves; instead, there is a morphological effect in the samples treated with the second dosage. The increase could also be due to the higher content of carbohydrates, which would correlate with the other FTIR regions. 

The diagnostic region provides the first evidence that Se-VF positively influences the biofortification of cabbage by increasing the intermolecular H-binding and the increased biosynthesis of lipids and aliphatic glucosinolates, and possibly selenoglucosinolates [104,105,106,107,108,109] and carbohydrates. 

A third FTIR fingerprint is linked to the bands around 1738 cm^−1^ and 1612 cm^−1^, which is characteristic of the uronic acid of pectin/hemicelluloses and is specific for methyl-esterified/protonated and unesterified/non-protonated carboxylate groups, based on which the degree of methylation and protonation (DMP, %) of pectin (galacturonan backbone) was calculated (Appendix A). Indeed, the absorption band around 1740 ± 50 cm^−1^ is notorious for the C=O vibration in various compounds, including esterified/protonated carbonyl COOR/H [78,93,118,159,168], which is used to evidence the accumulation of pectin in the *Brassica napus* cell wall [115]. The proportion of protonated carboxyl in our samples is probably not significant, as the bands at 873 cm^−1^ have very low intensity. The pectin is classified as high-methyl-esterified pectin, HMP (>50% DM), and low-methyl-esterified pectin, LMP (<50% DM) [169]. In our case, the DM of pectin, determined in Appendix A, suggests that pectin is highly esterified and probably partially protonated in the control samples Ci (50.68% DMP) and especially Co (61.89% DMP). The foliar treatments induced a significant demethylation/deprotonation of pectin in the outer leaves with almost 10% at the D1 dose and almost 30% at the 3× higher D2 dose. In the case of the inner leaves, the behavior is more complex, with an approx. 5% increase in DMP at the D1 dose and a significant 42% decrease in DMP at the 3× higher D2 dose. This shows that there are differences between the mechanism of the Se-VF effect on the inner and the outer leaves. Exogeneous selenium has been previously shown to enhance the activity of pectinesterase of *Brassica rapa*, which induces the demethylation of pectin [170]. This result correlates with the data obtained in our study, in which we observe an increase in demethylated pectin following the treatment with Se-VF, suggesting that Se present in the formulation is responsible for this demethylation. Glycine betaine acts as a methyl donor, and one would expect Se-VF to induce a higher DM compared to the control. This is observed only in the case of the inner leaves for the lower Se-VF dose (D1i). This means that at dose D1, the effects of GB might prevail over those of Se, and a dose of Se might not be enough to counteract the increased methylation induced by GB in the inner leaves. In the case of the outer leaves, the D1 dose of Se seems to be enough to induce a decrease in DM compared to the control. One possible factor contributing to this different behavior between inner and outer leaves could be related to the initial DM, which is higher in the outer leaves compared with the inner leaves. It is possible that the effect of Se depends on the initial DM, i.e., the higher the DM, the higher the effect on demethylation. It is not clear if and what effect the GB has on the outer leaves at D1 and D2 and the inner leaves at the D2 dose. Treatment without Se and treatment without vinasse should be performed in the future to evaluate the two effects separately. 

LMP has been generally associated with an increased rate of fermentation and higher production of short-chain fatty acids (SCFAs) by gut microbiota by increasing the proportion of *Lactobacillus* sp. [171,172]. LMP was also shown to stimulate the secretion of intestinal mucins and protect epithelial cells [173]. LMP was shown to have some additional health benefits, such as low type 1 diabetes (T1D) incidence and anti-inflammatory effects, which were recently reviewed in [174]. In general, the health benefits of LMP seem to surpass HMP. This indicates that our treatment might improve the health benefit of cabbage. The increase in the LMP fraction, which is more soluble in water than HMP [175], partially correlates with the increase in the soluble fibers of the inner leaves by the D2 dose following treatment. In the case of the D1 dose, the contribution to the soluble fibers must come from other compounds. 

A fourth FTIR fingerprint is the region of 1500–1200 cm^−1^. Here, the shifting and broadening of the band 1419(Ci)/1423(Co) cm^−1^ toward lower frequencies could be related to the demethylation of pectin; the band was previously reported for esterified pectin or pectin in salt form [134,176] and also for changes in cellulose crystallinity. The band with the frequency 1429 cm^−1^ is considered as a “crystalline” absorption band [177]. The shift of this band to lower frequencies and broadenings has been associated with a decrease in the crystallinity of cellulose [177,178]; but in a complex system such as leaf structure, the ratio between 1429 cm^−1^ and 879 cm^−1^ (in our case), which is characteristic to amorphous cellulose [178], should be more useful. The reduction, broadening, and shifting to a higher frequency of the band at 1221 cm^−1^ is correlated with the demethylation of pectin induced by Se-VF. 

The significant changes in DM of pectin could be partially correlated with the changes observed for the crystallinity index, as determined by XRD. When DM increases (for example, by 5% from Ci to D1i), Xc decreases, and when DM decreases (in the cases of D2i, D1o, and D2o compared to the controls), Xc increases. This suggests that the induced demethylation of pectin by Se-VF leads to slightly more crystalline pectin. This is in agreement with previous studies, which claimed that methylation induces a less crystalline organization of pectin [179]. An explanation for this behavior is that methylation disrupts the H bonds between pectin chains formed by -OH groups, which stabilize a more ordered and, therefore, a more crystalline structure. 

The fifth and last FTIR fingerprint can be considered as the increased absorption band of carbohydrates around 1030 ± 100 cm^−1^ following treatment with Se-VF, which suggests an increased development of structural chains of polysaccharides. Here, the ratio between the band at approx. 1100 cm^−1^, which is characteristic of crystalline cellulose, and the band at 879 cm^−1^ in our case, which is characteristic of amorphous cellulose [178], partially correlate with the crystallinity obtained by XRD (Table 3). One exception is for D1o, in which case the cellulose crystallinity apparently decreases, but the overall crystallinity increases. The overall crystallinity is not influenced only by changes in the cellulose crystallinity but also by changes in the DM of pectin, as mentioned above, and also by the content of lignin. Therefore, depending on the relative proportions of these components and their contribution to leaf crystallinity, the latter is the product of the equilibrium between the different components. The increase in D1o crystallinity compared to Co could be the effect of a reduced proportion of pectin/hemicelullose and lignin compared to cellulose, combined with the contribution from the approx. 10% reduction in DM of D1o compared to Co. As the soluble and insoluble fibers are not affected in D1o compared to Co, lignin and other amorphous structures are probably reduced in favor of cellulose. 

In the case of the inner leaves, the frequency ratios of D2i are much higher, but Xc is lower than Ci in Table 3. This discrepancy is explained by the significant increase in the soluble fibers, which are more amorphous than cellulose, which contributes to the reduction in the overall leaf crystallinity of Ci compared to Co. 

Considering the data discussed above, the take-home message is that leaf crystallinity involves several players, which are in equilibrium and cannot be explained based only on one component. The maximum peak in this region, in our case 1034/1018 cm^−1^, can be used to obtain a crude assessment of the esterification of cell walls by dividing the area of the 1738 cm^−1^ peak by its area. We obtained the same trend for the 1612 cm^−1^ band, which demonstrates the accuracy of our data and the usefulness of the method. 

Summarizing the FTIR fingerprints, it may be concluded that Se-VF stabilizes and increases pectins as unesterified galacturonic acids or carboxylates, especially in the inner leaves, and increases the lipids, aliphatic glucosinolates, and carbohydrates, especially cellulose Iα, as evidenced by the XRD analyses. The FTIR data can partially explain the contributions to the overall leaf crystallinity. 

The FTIR spectroscopic techniques applied to whole plant tissues were already used to reveal the different cultivation conditions. The application of a solid protein hydrolysate produced from the trimmings and shavings of bovine hides on maize leads to the modification of the FTIR spectra of leaves and roots [180]. These modifications were confirmed by solid-state cross-polarization magic angle spinning carbon-13 nuclear magnetic resonance, CP/MAS 13C–NMR, and high-resolution magic angle spinning nuclear magnetic resonance, HR-MAS NM. The FTIR spectra of roots, stems, leaves, and curds from *Brassica oleracea* L. var. *botrytis* revealed modifications produced by applications of nitrogen fertilizers and biochar [102]. Our approach was to combine FTIR spectroscopy with XRD spectroscopy and thermogravimetric analyses (TGAs) to better reveal the effects of the agrotechnical practices that increase plant tolerance to stressful conditions on the main components of the plant cell walls. As we mentioned already, a part of these components is important for the quality traits of leafy vegetables, e.g., as dietary fibers/healthy carbohydrates [77,181]. 

## 4. Materials and Methods

### 4.1. Materials

The experiment was performed on white cabbage (*Brassica oleracea* var. *capitata* L.), cv. Mirror F1. The culture was established by using transplants produced in greenhouse conditions at a density equivalent to 55,000 plants ha^−1^. The concentrated baker’s yeast vinasse used in this study was produced by Rompak (Pașcani, Romania). The following standards were used for analysis: micro-crystalline cellulose (MCC, Avicel PH-101, Merck-Sigma Aldrich, Saint Louis, MO, USA), pectin from citrus peels (Pct, galacturonic acid ≥ 74.0%, Merck-Sigma Aldrich, Saint Louis, MO, USA), and lignin (Lgn, Lignin alkali, kraft, low sulfonate, pH 10.5 at 3 wt%, Merck-Sigma Aldrich, Saint Louis, MO, USA). For pectin, cellulose, and lignin extraction from cabbage, NaOH pellets and 30% H_2_O_2_ from Cristal R Chim SRL Bucharest, Romania, 95–98% H_2_SO_4_ from Fluka Honeywell, Seelze, Germany were used. Cystine, and 2, 5-Bis(5-tert-butyl-2-benzo-oxazol-2-yl from Thermo Fisher Scientific, Waltham, MA, USA were used for the calibration curve of the elemental composition determination. 

### 4.2. Characterization of the Selenium–Vinasse Formulation (Se-VF)

The composition of the organo-mineral formulation Se-VF was determined, as reported by us for a similar foliar biostimulant tested on tomatoes [21]. Briefly, total Kjeldahl nitrogen (TKN) in Se-VF was determined using the protocol EN15478:2009 on a Behrotest InKjel P Digestion System (Behr Labor-Technik, Düsseldorf, Germany). Phosphorus was gravimetrically determined based on the reaction with ammonium quinoline molybdate according to the ISO 6598:1996 method. Potassium was determined using the method EN 15477:2009 by precipitation with sodium tetraphenylborate in a weak alkali environment. The concentration of microelements Se, Cu, Mn, Mg, Fe, Zn, and Mo was determined by inductively coupled plasma–optical emission spectrometry (ICP-OES) using an Optima 2100 DV Perkin Elmer equipment (Perkin Elmer, Waltham, MA, USA). 

Se-VF was designed as an organo-mineral foliar fertilizer enriched with selenium formulated with vinasse rich in glycine betaine (GB). The composition of Se-VF is presented in Table 4.

The Se-VF product was a stable, easily sprayable foliar formulation. The influence of Se-VF on the growth and structure of white cabbage (*Brassica oleracea*, Capitata Group) was studied at two different doses, encoded D1 and D2 (3×D1), in comparison with the control experiment, C. 

### 4.3. Cabbage Experimental Treatment 

The experimental plots were located on Stefan-cel-Mare, Călărași county, Romania, 44°25′28′′ N latitude, 27°38′24′′ E longitude, and 52 m altitude. The average values of multi-annual temperature, total precipitation, sunshine daily duration, and wind speed for this experimental site are 11.5 °C, 504 mm, 6.9 h, and, 3.5 ms^−1^, respectively.The soil is a calcaric kastanic chernozem with an average total selenium content in the upper soil horizon (0–20 cm) of 67 µg.kg^−1^. This selenium level is lower that almost 40% of the soil content considered unaffected by Se deficiencies [58]. The base fertilization was applied 5 days before cabbage crop establishment and consisted of inorganic fertilizers, which were applied in dose equivalent to N—160 kg ha^−1^, P—120 kg ha^−1^, and K—120 kg ha^−1^), and were applied 5 days before cabbage seedlings transplanting. An automated weather station (nMetos, Pessl Instruments, Weiz, Austria) located at 50 m from the experimental site was used to record the climatic condition on the experimental site. During the 2021 period of cabbage vegetation, these climatic conditions were characterized by higher monthly temperatures (+1.7 °C in May; +0.9 °C in June; +2.2 °C in July), lower monthly precipitations in May (−22.4 mm) and July (−17.7 mm), and higher monthly precipitations in June (+27.2 mm) than the average multi-annual. Irrigation was performed by a drip irrigation system installed on each row at 10 cm depth. 

The foliar treatment (Se-VF) was applied twice. The first one was at the end of June, after 4 weeks of growth of the transplanted seedlings when the plants were developed enough for the foliar spraying to be efficient. The second treatment and specific field measurements were made 2.5 weeks after the first treatment, in the middle of July. The treatments were performed by foliar spraying with a backup sprayer SGA 85 (Stilhl, Waiblingen, Germany), which is equivalent to a spraying solution per ha of 200 L. The applied doses were equivalent to 1 L/ha (D1, around 5 g Se salt) and 3 L/ha (D2, around 15 g Se salt). The experimental treatments, control, and D1 and D2 were performed in randomized block design with 3 replicates, each replicate containing 50 cabbage plants. 

### 4.4. Determination of Effects on Cabbage Physiology

Stomatal conductance and chlorophyll fluorescence in-field measurements were performed early in the morning. Stomatal conductance was determined by measuring the dynamic diffusion conductance of leaves using the portable AP4 porometer (DeltaT Devices Ltd., Cambridge, UK), in 5 replicates. Chlorophyll fluorescence was determined with the PAM2500 portable chlorophyll fluorometer (Heinz Walz GmbH, Effeltrich, Germany) equipped with LEDs for saturation pulses and blue and red actinic lights, a Leaf-Clip Holder 2030-B with an integrated micro-quantum sensor for recording the photosynthetic active radiation (PAR, in units of flux density μmol quanta/(m^2^·s)) and also temperature with a NiCr-Ni thermocouple. Hence, the measured PAR parameter is identical to PPFD (photosynthetic photon flux density). The effective photochemical quantum yield (Φ) of photosystem II was determined using the following equation: Φ_PSII_ = (F′′_m_—F_t_)/F′′_m_, where F′′_m_ represents the maximal chlorophyll fluorescence yield when photosystem II reaction centers are closed by a strong light pulse, and F_t_ represents the continuously recorded fluorescence, according to the manufacturer. In photochemistry, the symbol Φ is reserved for quantum yield, according to IUPAC Recommendations [182], and represents the number of defined events which occur per absorbed photon. 

### 4.5. Extraction of Pectin, Cellulose, and Lignin from Cabbage

In order to evaluate the pectin, cellulose, and lignin particularities in cabbage, a cascade extraction was proposed based on adapted individual protocols reported in the literature. The first step consisted of a 2 h hydrothermal extraction (HTE) at 130 °C of 80 g fresh ground cabbage with 50 mL of H_2_O. Using a Parr compact reactor 5500 Series, the protocol for pectin extraction was adapted after [183]. The resulting liquid phase contains pectin and other hydrosoluble compounds; therefore, 96% ethanol was added for pectin precipitation and further separation [183]. The solid phase from HTE contains mainly cellulose and lignin; therefore, the bleaching protocol “dancing with a dragon” was adapted [184] by using a 5% NaOH solution for 3 h under sonication at 70 °C, followed by the addition of 10% H_2_O_2_ and H_2_O_2_and for another 3 h of effervescent bleaching. After bleaching, the solution was left to cool down and was further filtrated through a Whatman No. 1 cellulose filter, washed three times with double-distilled water under sonication for 1h each step at 70 °C, and finally collected, dried, weighed, and analyzed. The liquid alkaline filtrate contains solubilized lignin, which was precipitated with 95–98% concentrated H_2_SO_4_ [185], was further separated, washed three times at 70 °C under sonication, and was filtered, dried, weighed, and analyzed. 

### 4.6. Thermo-Gravimetrical Analysis

Thermo-gravimetrical analyses (TGAs) were performed on dried samples of the control and treated cultivars with the help of Q5000IR equipment (TA Instruments, New Castle, DE, USA) by adding 1–4 mg in each 100 µL platinum pan. The TGA method was performed by heating the samples from 20 °C to 525 °C with a temperature ramp of 10 °C/min under nitrogen (99.99%) purged with 40 mL/min. At 525 °C, the purge gas was switched to synthetic air (99.99%) in hi-res mode and kept isothermal for 30 min to determine the ash content. An upper temperature limit of 525 °C was chosen to be similar to the ash determination temperature suggested on the Megazyme Total Dietary Fiber assay procedure K-TDFR-100A/K-TDFR-200A 04/17.

### 4.7. Spectroscopic Analysis

#### 4.7.1. X-ray Diffraction Analysis

X-ray diffraction (XRD) data were obtained using a Rigaku SmartLab diffractometer (Rigaku Corporation, Tokyo, Japan) with Cu_Kα1_ (λ = 1.54059 Å) radiation working on wide-angle X-ray diffraction measurements in the 2θ interval 5–90° with 0.02° resolution and a scan speed of 4°/min. The tube voltage was set at 40 kV and the emission current at 200 mA. Measurements were made in parallel beam configuration on powder samples obtained after drying, grinding, and sieving. Rigaku-PDXL 2.7.2.0 software was used for smoothing the diffractograms using the B-Spline model with Chi = 1 and for background subtraction, peak deconvolution, and identification. OriginPro software version 9 was used for the final graphical representation. The crystallinity degree (Xc, %) was determined using the Rigaku-PDXL 2.7.2.0 software as the ratio between the area of crystalline peaks and the total peaks area (of the crystalline and amorphous peaks). 

#### 4.7.2. ATR-FTIR Spectroscopy

For each experiment, relevant plant material was taken from the inner and outer leaves of cabbage and encoded with the letters “i” and “o”, respectively. Plant material was dried in an oven at 55 °C for 2 days and was then crushed and sifted through a 0.5 mm sieve. The fraction < 0.5 mm was further used for analyses. Fourier-transform infrared (FTIR) spectra were recorded with the attenuated total reflectance (ATR) technique on an IRTracer-100 FTIR (Shimadzu, Kyoto, Japan) in wavenumbers ranging from 4000 to 400 cm^−1^ by the accumulation of 45 spectra at a resolution of 4 cm^−1^. Commercial MCC, pectin from citrus peels, and lignin alkali were used as FTIR standard macromolecules. The exported files were processed and graphically overlaid using OriginPro software version 9 from OriginLab Corporation (Northampton, MA, USA). 

### 4.8. Content of Soluble and Insoluble Fibers

The soluble and insoluble fibers were determined using the commercial Total Dietary Fiber Assay kit from Megazyme (Bray, Ireland), following the kit protocol. Total protein was determined by the Kjeldahl method and subtracted, together with the ash determined with TGAs (ResAir), from the total insoluble fraction determined with the Megazyme kit, to obtain the insoluble dietary fiber. The ratio between the soluble fraction and insoluble fraction was calculated and compared to the crystallinity determined by XRD. 

### 4.9. Elemental Analysis

The determination of N, C, and H content was carried out with a FlashSmart Elemental Analyzer (Thermo Fisher Scientific, Waltham, MA, USA) equipped with a thermal conductivity detector (TCD). The samples were burned at 950 °C in an oxygen atmosphere (99.999% purity). The equipment calibration was performed using a reference material, i.e., 2,5-Bis(5-tert-butyl-2-benzo-oxazol-2-yl (N = 6.51 ± 0.09%; C = 72.52 ± 0.22%; H = 6.09 ± 0.08%). The calibration curve ranges were 0.008–0.172 mg for nitrogen (R^2^ = 0.9997), 0.089–1.914 mg for carbon (R2 = 0.9996), and 0.007–0.161 mg for hydrogen (R2 = 0.9995). The calibration curves were checked with cystine (N = 11.66 ± 0.16%, C = 29.98 ± 0.28%, H = 5.03 ± 0.13%). 

### 4.10. Statistical Analysis

Statistical analysis was performed using IBM SPSS 26 software. 

## 5. Conclusions

The foliar-tested Se-VF product applied on white cabbage grown in a drought area induced a faster growth metabolism, with faster pectin and glucosinolates accumulation, together with a biofortified structure rich in cellulose Iα and stabilized by an increased number of H-bonds compared to the control. The TGA suggests imagining cabbage as a molecular factory with nine thermo-regions, in which the movement of the molecular building blocks is accelerated by foliar treatments with Se-VF toward soluble fibers, pectin, cellulose, and biomass accumulation, with lignin in the role of a molecular conveyor-belt. The TGA confirmed the pectin and cellulose accumulation by weight losses in their corresponding thermo-regions, and also cabbage biofortification with minerals through an increase in the ash residue. The XRD fingerprint shows, for the first time, the predominance of cellulose Iα together with pectin in cabbage. The FTIR fingerprint in the region 1738 cm^−1^ and 1612 cm^−1^ showed the stabilization of pectin in an unesterified, demethylated form. The XRD and FTIR fingerprints correlate with each other in terms of leaf crystallinity. As a final conclusion, we showed that physicochemical methods such as FTIR, XRD, and the TGA can be used to fingerprint compositional, structural, and physiological changes induced by plant biostimulants based on Se and vinasse. The results of this study demonstrate that spectroscopic techniques and thermogravimetric analysis have the potential to reveal structural changes in the leaves of plants treated with plant biostimulants. Applying these new methods of characterization will complement biochemical, physiological, and agrotechnical methods for understanding the agricultural function and action mechanisms of plant biostimulants. Based on this better understanding of the plant biostimulant action on cultivated plants, improved technologies that will increase the reproducibility and efficiency of these 21st-century agricultural inputs could be developed. 

## Figures and Tables

**Figure 1 plants-12-03016-f001:**
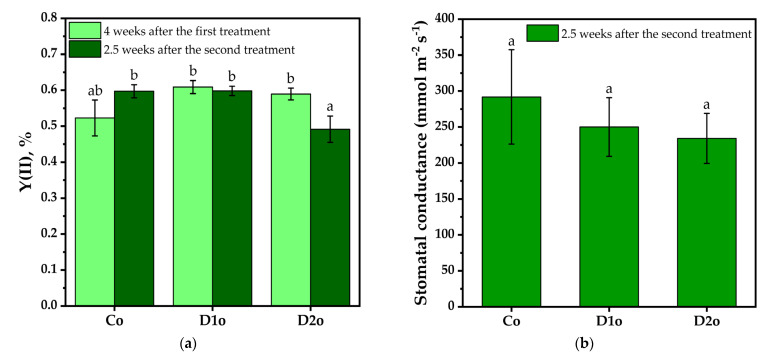
Photosynthetic activity of cabbage outer leaves (o) at 4 weeks after the first treatment and 2.5 weeks after the second treatment: (**a**) chlorophyll fluorescence, (**b**) stomatal conductance; Co - control outer leaves, D1o - dose 1 (1 L/ha) outer leaves, D2o - dose 2 (3 L/ha) outer leaves (±error bars, *n* = 6 for Y(II), and n = 5 for stomatal conductance; significance level α = 0.05. Different letters indicate significant differences between samples; double letters indicate statistical similarities between samples).

**Figure 2 plants-12-03016-f002:**
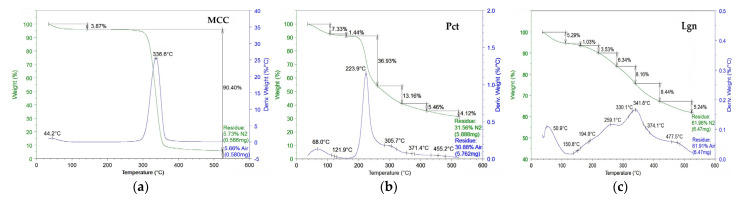
Thermo-gravimetrical analyses of (**a**) MCC- microcrystalline cellulose; (**b**) Pct-pectin; (**c**) Lgn-lignin; (**d**) Ci-control cabbage inner leaves; (**e**) Co - control cabbage outer leaves; (**f**) D1i - dosage 1 on inner leaves; (**g**) D1o - dosage 1 on outer leaves; (**h**) D2i - dosage 2 on inner leaves; (**i**) D2o—dosage 2 on outer leaves.

**Figure 3 plants-12-03016-f003:**
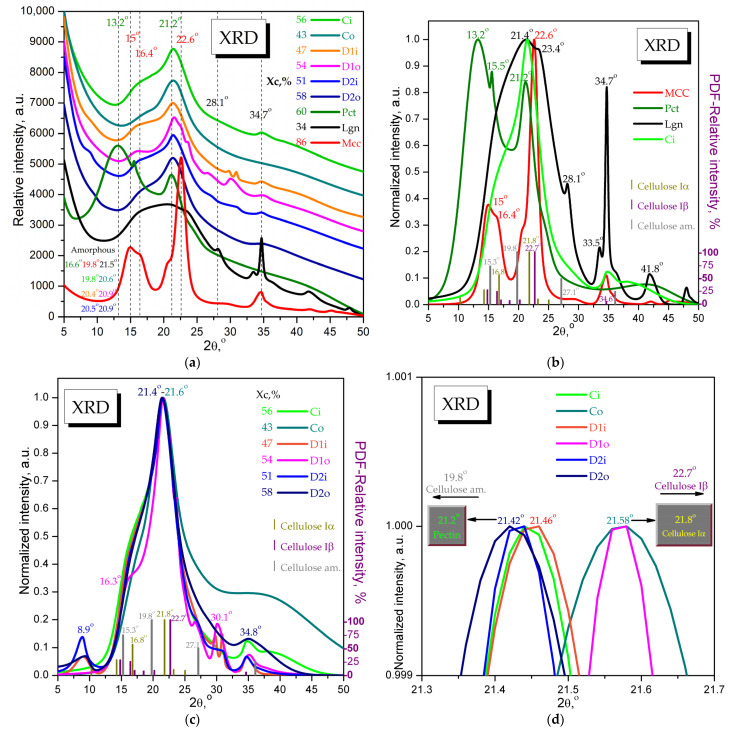
X-ray diffractograms of cabbage samples and standards: C—cabbage control (i,o); D—dosage (1,2); i—inner leaves; o—outer leaves; MCC—microcrystalline cellulose; Pct—pectin; Lgn—lignin. (**a**) Smoothed B-spline diffractograms translated and overlaid to evidence the convolution of MCC, Pct, and Lgn in cabbage samples; (**b**) smoothed, background-subtracted and normalized diffractograms of MCC, Pct, Lgn, and Ci, along with the typical main peaks of cellulose Iα, Iβ, and amorphous cellulose; (**c**) normalized cabbage diffractograms together with the main peaks of cellulose Iα, Iβ, amorphous cellulose, and methylcellulose; (**d**) close-up of the main diffraction peak in cabbage samples, showing a small shift toward pectin’s peak 21.2° of inner leaves and biostimulant-treated D2o leaves. Crystallinity degrees (Xc, %) are presented in (**a**,**c**), while all the amorphous deconvoluted peaks are presented as a small table in the left-down corner of (**a**).

**Figure 4 plants-12-03016-f004:**
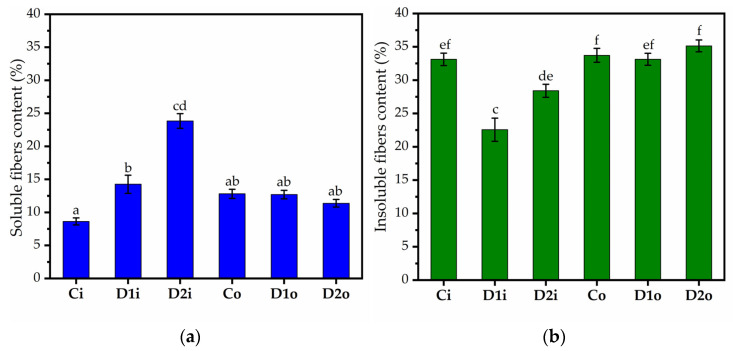
Fiber type and content in cabbage leaves: (**a**) soluble fibers; (**b**) insoluble fibers Ci—control cabbage inner leaves; Co—control cabbage outer leaves; D1i—dosage 1 (1 L/ha) on inner leaves; D1o—dosage 1 on outer leaves; D2i—dosage 2 (3 L/ha) on inner leaves; D2o—dosage 2 on outer leaves. (±error bars, n = 3, significance level α = 0.05, different letters indicate significant differences between samples).

**Figure 5 plants-12-03016-f005:**
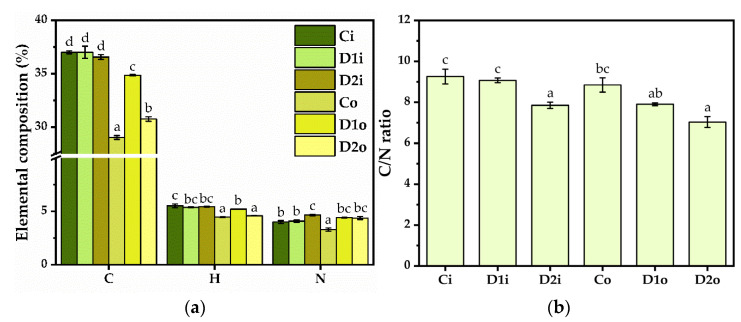
Elemental analysis and protein content after 2.5 weeks following the second treatment: (**a**) carbon (C), hydrogen (H), nitrogen (N) content, (**b**) C/N ratio; (±standard deviation, n = 3, α = 0.05); Ci - control cabbage inner leaves; Co - control cabbage outer leaves; D1i - dosage 1 (1 L/ha) on inner leaves; D1o - dosage 1 on outer leaves; D2i - dosage 2 (3 L/ha) on inner leaves; D2o - dosage 2 on outer leaves. (±error bars, n = 3, significance level α = 0.05, different letters indicate significant differences between samples, double letters indicate statistical similarities between samples).

**Figure 6 plants-12-03016-f006:**
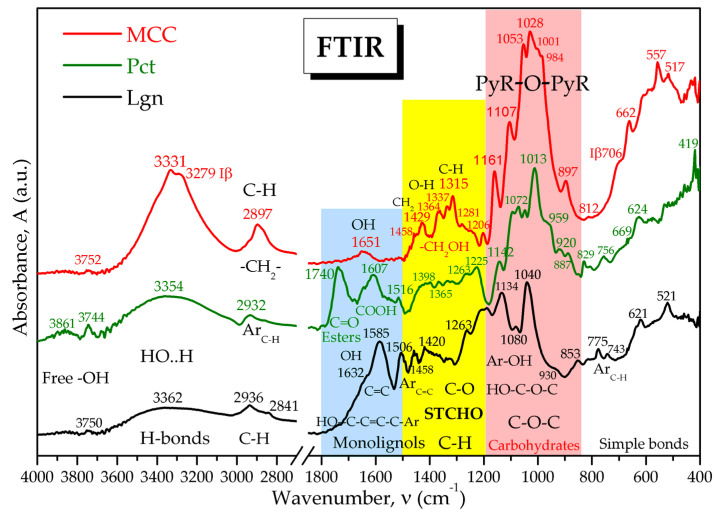
FTIR spectra of commercial microcrystalline cellulose (MCC), pectin (Pct), and lignin (Lgn).

**Figure 7 plants-12-03016-f007:**
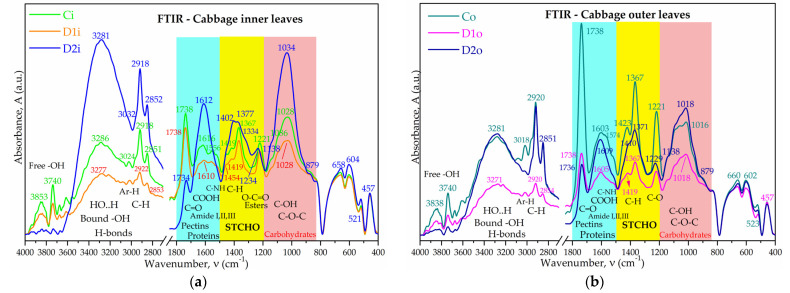
FTIR spectra of cabbage samples foliar-treated with two dosages of Se-BNF, D1 - 1L/ha (code 1), and D2 - 3L/ha (code 2) in comparison with the control inner leaves (Ci) and outer leaves (Co): (**a**) cabbage inner leaves (Ci, D1i, and D2i); (**b**) cabbage outer leaves (Co, D1o, and D2o).

**Table 1 plants-12-03016-t001:** Thermo-gravimetrical analyses of MCC, Pct, Lgn, the controls, and treated inner and outer leaves of cabbage.

Sample	T_1_ (°C), WL_1_ (%) iwH_2_O and VOBs 25–105 °C	T_2_ (°C), WL_2_ (%) PF and AA and sH_2_O 105–160 °C	T_3_ (°C), WL_3_ (%) prot and sol.fib. 160–220 °C	T_4_ (°C), WL_4_ (%) sol.fib./pectin 220–260 °C	T_5_ (°C), WL_5_ (%) Cellulose 260–340 °C	T_6_ (°C), WL_6_ (%) Lignin 340–420 °C	T_7_ (°C), WL_7_ (%) OxBiochar 420–525 °C	ResN_2_, w% 525 °C	ResAir, w% 525 °C
Mcc	44.2, 3.87%	-	-	-	336.6, 90.40%	-	-	5.73%	5.66%
Pct	68.0, 7.33%	121.9, 1.44%	-	223.9, 36.93%	305.7, 13.16%	371.4, 5.46%	455.2, 4.12%	31.56%	30.88%
Lgn	50.9, 5.29%	150.8, 1.03%	194.9, 3.53%	259.1, 6.34%	330.1, 8.16%	341.8, 8.44%	477.5, 5.24%	61.97%	61.91%
Ci	74.2, 4.06%	136.2, 6.26%	194.4, 11.99%	237.7, 8.38%	300.3, 19.22%	347.6, 8.20%	454.2, 5.80%	36.09%	11.05%
Co	61.7, 4.37%	138.7, 6.08%	189.1, 10.66%	239.1, 9.68%	298.3, 21.03%	367.5, 7.97%	463.1, 4.87%	35.34%	12.07%
D1i	59.8, 4.57%	135.1, 6.35%	180.9, 8.98%	238.6, 11.06%	297.5, 20.34%	361.2, 8.60%	440.6, 5.38%	34.72%	11.10%
D1o	50.7, 4.98%	131.3, 4.43%	185.8, 7.99%	240.2, 12.85%	288.2, 21.26%	358.7, 9.47%	446.8, 4.82%	34.20%	15.14%
D2i	59.7, 4.21%	137.5, 7.73%	187.9, 9.19%	240.0, 9.59%	299.5, 21.29%	349.5, 7.39%	445.0, 5.13%	35.47%	12.92%
D2o	50.3, 5.29%	137.8, 3.77%	189.7, 7.02%	238.5, 12.55%	289.1, 24.33%	354.8, 7.65%	453.5, 4.65%	34.74%	16.08%

Notations: iwH_2_O—interstitial and weakly bound water; VOBs—volatile organic biocompounds; PFs—polyphenols; AAs—aminoacids; sH_2_O—strongly bound water; prot.—proteins; sol.fib.—soluble fibers; C—cabbage control; D—dosage; i—inner leaves; o—outer leaves; MCC—microcrystalline cellulose; Pct—pectin; Lgn—lignin.

**Table 2 plants-12-03016-t002:** Dietary soluble and insoluble fibers (%) and their ratio and correlation with the crystallinity determined by XRD; Ci—control cabbage inner leaves; Co—control cabbage outer leaves; D1i—dosage 1 (1 L/ha) on inner leaves; D1o—dosage 1 on outer leaves; D2i—dosage 2 (3 L/ha) on inner leaves; D2o—dosage 2 on outer leaves. (±standard error).

Sample	% Soluble Fiber Fraction	% Insoluble Fiber Fraction	Ratio (Soluble/Insoluble)	Crystallinity % (XRD)
Ci	8.65 ± 1.04	33.13 ± 1.83	0.261	56
D1i	14.27 ± 2.69	22.57 ± 3.39	0.632	47
D2i	23.85 ± 2.15	28.42 ± 1.92	0.839	51
Co	12.83 ± 1.34	33.73 ± 2.04	0.381	43
D1o	12.72 ± 1.23	33.13 ± 1.74	0.384	54
D2o	11.40 ± 1.13	35.14 ± 1.74	0.324	58

**Table 3 plants-12-03016-t003:** Comparison between variations in cellulose crystallinity, as determined by FTIR, and the XRD crystallinity.

Sample	Freq 1 (1100 cm^−1^)	Freq 2 (879 cm^−1^)	Ratio (Freq 1/Freq 2)	Crystallinity % (XRD)
Ci	0.076	0.049	1.532	56
D1i	0.064	0.047	1.355	47
D2i	0.106	0.047	2.242	51
Co	0.100	0.055	1.823	43
D1o	0.067	0.050	1.353	54
D2o	0.104	0.054	1.935	58
**Sample**	**Freq 3 (1419 cm^−1^)**	**Freq 2 (879 cm^−1^)**	**Ratio (Freq 3/Freq 2)**	**Crystallinity % (XRD)**
Ci	0.067	0.049	1.367	56
D1i	0.059	0.047	1.258	47
D2i	0.089	0.0470	1.866	51
Co	0.105	0.055	1.904	43
D1o	0.061	0.049	1.222	54
D2o	0.089	0.054	1.656	58

**Table 4 plants-12-03016-t004:** Characteristics of the tested Se–vinasse formulation (Se-VF).

Parameter	Units	Se-BNF *
Total nitrogen, N	g/L	105.7
Total phosphorus, P	g/L, as P_2_O_5_	36.6
Water-soluble potassium, K	g/L, as K_2_O	40.2
Glycine betaine from vinasse	g/L	16.7
Iron, Fe	g/L	0.40
Manganese, Mn	g/L	0.36
Copper, Cu	g/L	0.18
Boron, B	g/L	0.12
Magnesium, Mg	g/L	0.11
Zinc, Zn	g/L	0.06
Molybdenum, Mo	g/L	0.02
Selenium, Se	g/L	0.06
pH	pH unit	6.73
Density	kg/L	1.187

* Aspect: brown homogeneous liquid, slightly viscous.

## Data Availability

Data are available upon request from the authors.

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
