# Peer review of "Spectroscopic Analyses Highlight Plant Biostimulant Effects of Baker’s Yeast Vinasse and Selenium on Cabbage through Foliar Fertilization"

_plants, 2023, doi:10.3390/plants12163016_

Round 1
Reviewer 1 Report
Dear Editors,
Thank you so much for choosing me as a reviewer of the manuscript plants-2497954 entitled: “Spectroscopic Analyses Highlight Plant Biostimulant Effects of Baker’s Yeast Vinasse and Selenium on Cabbage through Foliar Fertilization”. I hope that my comments will help Authors to improve their manuscript. Detailed remarks concerning the manuscript:
1. A clear scientific hypothesis together with the answer to the question stated as scientific hypothesis should be given.
2. It is not recommended to use as key words the words or phrases used in the title of the manuscript. Please do needed changes.
3. Please pay the attention to the number of the sections and subsections see Results 2.1. Preparation and characterization of Se-VF; 2.2 Treated plants physiologycal activity; 3.3. Thermo-gravimetrical analyses - TGA; 2.4. X-Ray diffraction analyses - XRD.
4. All the figures and tables should be clear for the reader without the referring to the text of the manuscript. Please give the explanations where needed.
5. The bibliography should be verified due to editorial mistakes. For example: there is no full bibliographic data for the references: 35. Yousaf, M.; Bashir, S.; Raza, H.; Shah, A.N.; Iqbal, J.; Arif, M.; Bukhari, M.A.; Muhammad, S.; Hashim, S.; Alkahtani, J., et al. Role of nitrogen and magnesium for growth, yield and nutritional quality of radish. and 44. Campos-Soriano, L.; Bundó, M.; Bach-Pages, M.; Chiang, S.F.; Chiou, T.J.; San Segundo, B.A.-O. Phosphate excess increases susceptibility to pathogen infection in rice. Please go through the whole bibliography in order to correct all editorial mistakes. Please do changes for the reference 128. Daxenbichler, M.E.; Spencer, G.F.; Carlson, D.G.; Rose, G.B.; Brinker, A.M.; Powell, R.G. GLUCOSINOLATE COMPOSITION OF SEEDS FROM 297 SPECIES OF WILD PLANTS. Phytochemistry 1991, 30, 2623-2638, doi:10.1016/0031-9422(91)85112-d. - why the title of the manuscript is italicized?
6. All Latin names of the species should be italicized. Please go through the whole manuscript and do needed changes.
7. the clear practical application of the studies should be given.
Author Response
Thank you for your comments which significantly contributed to the improvement of our manuscript.
Point 1. A clear scientific hypothesis together with the answer to the question stated as scientific hypothesis should be given.
Response 1. Thank you for your recommendation. The scientific hypothesis of this study considers plant cell walls and their components as an indicator of plant response to cultivation conditions, including those responses to biotic and abiotic stress. The main aim of this study is to reveal such integrated response at the level of the plant cell wall by characterizing the whole plant leaf using spectroscopic techniques and thermogravimetric analysis (TGA). We applied a fertilizing product, including a foliar fertilizer formula and plant biostimulants, as a tool to differentiate plant response to stress. Our study demonstrates that spectroscopic techniques and TGA spotlight structural changes in the leaves of a cultivated plant on which a new agrotechnical sequence, e.g., a fertilizing product, is applied. We added and highlighted these aspects at the beginning of the Abstract and in the last paragraphs of the Introduction Section.
Point 2. It is not recommended to use as key words the words or phrases used in the title of the manuscript. Please do needed changes.
Response 2. Thank you for your recommendation. We revised the key words and changed with: FTIR spectroscopy, XRD, thermogravimetric analysis plant cell wall components, plant response, soluble and insoluble fibers ,.
Point 3. Please pay the attention to the number of the sections and subsections see Results 2.1. Preparation and characterization of Se-VF; 2.2 Treated plants physiologycal activity; 3.3. Thermo-gravimetrical analyses - TGA; 2.4. X-Ray diffraction analyses - XRD.
Response 3. Thank you for your recommendation. The numbers of the sections were revised according to your requirement.
Point 4. All the figures and tables should be clear for the reader without the referring to the text of the manuscript. Please give the explanations where needed.
Response 4. The captions of the figures were rechecked and detailed where necessary.
Point 5. The bibliography should be verified due to editorial mistakes. For example: there is no full bibliographic data for the references: 35. Yousaf, M.; Bashir, S.; Raza, H.; Shah, A.N.; Iqbal, J.; Arif, M.; Bukhari, M.A.; Muhammad, S.; Hashim, S.; Alkahtani, J., et al. Role of nitrogen and magnesium for growth, yield and nutritional quality of radish. and 44. Campos-Soriano, L.; Bundó, M.; Bach-Pages, M.; Chiang, S.F.; Chiou, T.J.; San Segundo, B.A.-O. Phosphate excess increases susceptibility to pathogen infection in rice. Please go through the whole bibliography in order to correct all editorial mistakes. Please do changes for the reference 128. Daxenbichler, M.E.; Spencer, G.F.; Carlson, D.G.; Rose, G.B.; Brinker, A.M.; Powell, R.G. GLUCOSINOLATE COMPOSITION OF SEEDS FROM 297 SPECIES OF WILD PLANTS. Phytochemistry 1991, 30, 2623-2638, doi:10.1016/0031-9422(91)85112-d. - why the title of the manuscript is italicized?
Reponse 5. Thank you. It was a mistake on the initial data introduced in the references software. The bibliography style was rechecked for consistency.
Point 6. All Latin names of the species should be italicized. Please go through the whole manuscript and do needed changes.
Response 6. All Latin names were revised as suggested.
Point 7. The clear practical application of the studies should be given.
Response 7. The practical application was highlighted in the Conclusions section. The results of this study demonstrate that spectroscopic techniques and thermogravimetric analysis have the potential to reveal structural changes in the leaves of the plant treated with plant biostimulants. Applying these new characterization methods will complement biochemical, morpho-physiological, and agronomic methods for understanding the agricultural function and action mechanisms of the newly developed agrotechnical practices. Based on this better understanding, fine-tuning of the new agrotechnical technologies could be developed.
Reviewer 2 Report
I do not understand the aim of this study. It seems that the authors applied biostimulants to cabbage, and then did lots of different techniques for no apparent reason. The results are almost impossible to read, and the discussion is both too long and includes many far-fetched and/or impossible theories. The authors seem insufficiently acquainted with cell wall structure and composition and with plant structure.
Some specific comments:
L38 “Plant biostimulants are a class of agricultural inputs”- what do you mean by “inputs”?
L41“Selenium (Se) is a plant-beneficial nutrient, with a narrow 41 physiological window”- what do you mean by “narrow physiological window”? Please rewrite this paragraph to make it clear for the reader
L49 –“Selenium metabolism involves methylation and deplete S-Adenosylmethionine metabolic pool”- please change to “depletes”
Please explain what is “vinasse”- most readers won’t understand the term
L88 “excess potassium decreases the plant development”- what do you mean by “plant development”?
Introduction must be improved. It details the composition of the biostimulant- which should be in M&M section. The background is not clear.
Results- the description of formulation belongs to M&M. Everything between lines 100-109.
Figure 1 and 2- I do not see controls. It’s very unclear what treatments are shown
Figures 3 and 4, table 3- I cannot understand what the different colours and letters mean. The figures are very difficult to read.
In figure 6 legend it’s written there are 2 treatments and control (C)- but on the figure there are either Ci or Co. The reader cannot guess each time what the letters mean.
Discussion
The authors claim their treatment increases water use efficiency based on simple stomatal conductivity and photosynthesis measurements. This conclusion cannot be drawn at all, WUE must be calculated properly according to the literature.
The following paragraph makes no sense “The thermo-gravimetrical analyses suggest looking at plants as buildings under construction in which lignin, with weight fragments in all thermo-regions, can be seen as the molecular conveyor-belt of the plant. The bonds between cellulose wall under construction and the lignin scaffoldings, also known as ligno-cellulosic structures, lead to higher onset temperatures in inner leaves. In the outer leaves, the cellulose structure is finished, with higher cellulose percents, and the peripheral lignin scaffoldings are partially disassembled and further converted to plants' needs or are found as fragments into the complex structure of lignin. The weight loss variations in the 9-proposed thermo-regions suggest an accelerated development of the molecular building blocks by foliar treatments with Se-474 VF. TGA also confirmed the FTIR and XRD results on pectin and cellulose Iα accumulation in the particular T4 and T5 thermo-regions presented in Table 2.”
Plants are not made from lignin and cellulose, certainly not herbaceous leafy plant as cabbage! What peripheral lignin scaffoldings exactly? It seems the authors are not familiar with plant structure, or cell wall structure, or understand there are many cell types and cell wall types….
Assigning special properties to molecules at different T regions is far fetched in my opinion. There’s no proof whatsoever for any influence on stress resistance at all.
“All inner leaves have higher WL than outer leaves, which 499 suggests that the biochemical processes that synthesize soluble fibers and proteins are 500 more intense in the young leaves, and it can also be seen as an intermediary biosynthesis 501 step in the development of the plant, i.e., soluble fibers transformation into insoluble, 502 structural fibers like cellulose and lignin”
- again, completely far fetched. No proof for that, and the rest of the data does not support your hypothesis. Please show a difference in protein levels.
“As proposed, if we see the 508 biosynthesis of soluble fibers as an intermediary step in plant development, it means that 509 our Se-BNF accelerates the plant metabolism through higher biosynthesis of volatile 510 metabolomics, polyphenols, amino acids and peptides (T1-T2), faster conversion into ther-511 molabile fibers such as proteins and oligosaccharides (T3) and further into more thermo-512 stable soluble fibers, insoluble and structural fibers, fact evidenced by the lower content 513 in soluble fibers in the fertilized samples in region T3, and confirmed in the next two 514 thermo-regions T4 and T5 by higher content in hemicelluloses, pectins and cellulose in 515 treated samples.”
– the authors propose pectins (or as they call them – soluble fibers)- are intermediate step?????? It seems the authors never read a book on cell wall composition. Pectins are a constant part of any primary cell wall, while in cabbage those are the majority of cell walls.
Some mistakes with plural and singular, and several bad phrasings
Author Response
Point 1. I do not understand the aim of this study. It seems that the authors applied biostimulants to cabbage, and then did lots of different techniques for no apparent reason. The results are almost impossible to read, and the discussion is both too long and includes many far-fetched and/or impossible theories. The authors seem insufficiently acquainted with cell wall structure and composition and with plant structure.
Response 1. The main aim, hypothesis, and practical applications were highlighted more clearly in the Abstract, Introduction and Conclusions Sections. The Results section was clarified with the help of the Editor and all three Reviewers. Thank you again for all the suggestions. Specific discussions and references about the plant cell wall structure and composition were added.
Point 2. Some specific comments: L38 “Plant biostimulants are a class of agricultural inputs”- what do you mean by “inputs”?
Response 2. The “agricultural inputs” is a generic term that includes resources used for plant cultivation and animal husbandry, such as seeds, fertilizers, plant protection products, animal feed, veterinary drugs and dietary supplements, equipment and machineries, fuels and energy, labor force. A significant number of articles refers to plant biostimulants as “agricultural inputs” – here we cited randomly three of them: “there is growing scientific evidence supporting the use of biostimulants as agricultural inputs on diverse plant species…” (Calvo, P., Nelson, L. & Kloepper, J.W., 2014, Agricultural uses of plant biostimulants. Plant Soil 383, 3–41 – a Marschner Review with 976 citations on Clarivate Web of Science); “PBs including natural substances and microbial inoculants appear as a novel and potential category of agricultural inputs…” (Rouphael, Y., & Colla, G. (2020). Biostimulants in agriculture. Frontiers in plant science, 11, 40 – an editorial with 257 citations on Clarivate Web of Science); “Where biostimulants initially differ from other agricultural inputs is in the versatility of individual products regarding the desired response.” (Sible, C. N., Seebauer, J. R., & Below, F. E. (2021). Plant biostimulants: A categorical review, their implications for row crop production, and relation to soil health indicators. Agronomy, 11(7), 1297).
Point 3. L41“Selenium (Se) is a plant-beneficial nutrient, with a narrow physiological window”- what do you mean by “narrow physiological window”? Please rewrite this paragraph to make it clear for the reader.
Response 3. Thank you for your recommendation. We briefly explained the "narrow physiological window" as that "meaning that Se features beneficial effects in a concentration range lower than one magnitude order and dependent on the speciation (oxidation state and adsorbed forms)" (lines 60-62). We added several references that reader can refer.
Point 4. L49 –“Selenium metabolism involves methylation and deplete S-Adenosylmethionine metabolic pool”- please change to “depletes”.
Response 4. Performed as suggested (line 76).
Point 5. Please explain what is “vinasse”- most readers won’t understand the term.
Reponse 5. The term "vinasse" was in more detail explained and references provided (lines 90-92).
Point 6. L88 “excess potassium decreases the plant development”- what do you mean by “plant development”?
Response 6. Thank you. We modified to avoid confusion. The reference was updated, being displaced, and "plant development" was replaced with "root development and seedlings growth" (line 121).
Point 7. Introduction must be improved. It details the composition of the biostimulant- which should be in M&M section. The background is not clear.
Response 7. The Introduction was updated and the details regarding the composition of the biostimulant were moved to M&M, as suggested. We made efforts to clarify the background.
Point 8. Results- the description of formulation belongs to M&M. Everything between lines 100-109.
Response 8. Performed as suggested. The sub-section "2.1 Preparation and characterization of Se-VF" was moved in M&M and merged with the sub-section "4.2. Characterization of the selenium-vinasse formulation (Se-VF)".
Point 9. Figure 1 and 2- I do not see controls. It’s very unclear what treatments are shown.
Response 9. Thank you. Indeed, the control in Fig.1 should be "Co" instead of "C", also "D1o" and "D2o" instead of "D1" and "D2" since all photosynthesis results refer to the outer leaves. We improved also the figure caption. In Fig. 2 they are correct as "Ci, Co…".
Point 10. Figures 3 and 4, table 3- I cannot understand what the different colours and letters mean. The figures are very difficult to read.
Response 10. For Fig. 3 and 4, and Table 3 (new number as Table 2) additional caption details were added regarding the meaning of D1i, D1o … letters and also for the statistical SPSS letters in Fig.4: "different letters indicate significant differences between samples". In Figure 3, we used different colors to differentiate between samples. In Figure 4, "Blue" color represents soluble fibers and "Green" color represents insoluble fibers. We modified the y axis title to avoid confusion. We used different colors to be easier to follow and to look better, but we can also change to the same color if necessary. Additional) details were added in the caption of Fig. 4.
Point 11. In figure 6 legend it’s written there are 2 treatments and control (C)- but on the figure there are either Ci or Co. The reader cannot guess each time what the letters mean.
Response 11. Additional details were provided to clarify the caption of Fig. 6.
Point 12. Discussion. The authors claim their treatment increases water use efficiency based on simple stomatal conductivity and photosynthesis measurements. This conclusion cannot be drawn at all, WUE must be calculated properly according to the literature.
Response 12. Thank you for your comment. We modified, trying to explain better that Stomatal conductance is just an indicator of water use efficiency. This indicator refers to the equilibrium between transpiration and CO2 uptake - Lawson, T., & Blatt, M. R. (2014). The stomatal size, speed, and responsiveness impact the photosynthesis and water use efficiency (Plant physiology, 164(4), 1556-1570). Several studies used the stomatal conductance determined by leaf porometer as indicator of water use efficiency in plants from Brassicaceae family: Xu, B., Long, Y., Feng, X., Zhu, X., Sai, N., Chirkova, L., ... & Gilliham, M. (2021). GABA signalling modulates stomatal opening to enhance plant water use efficiency and drought resilience. Nature Communications, 12(1), 1952; Saja-Garbarz, D., Ostrowska, A., Kaczanowska, K., & Janowiak, F. (2021). Accumulation of Silicon and Changes in Water Balance under Drought Stress in Brassica napus var. napus L. Plants, 10(2), 280; Chrysargyris, A., Prasad, M., Kavanagh, A., & Tzortzakis, N. (2019). Biochar type and ratio as a peat additive/partial peat replacement in growing media for cabbage seedling production. Agronomy, 9(11), 693; Faralli, M., Grove, I. G., Hare, M. C., Boyle, R. D., Williams, K. S., Corke, F. M., & Kettlewell, P. S. (2016). Canopy application of film antitranspirants over the reproductive phase enhances yield and yield-related physiological traits of water-stressed oilseed rape (Brassica napus). Crop and Pasture Science, 67(7), 751-765.
Point 13. The following paragraph makes no sense “The thermo-gravimetrical analyses suggest looking at plants as buildings under construction in which lignin, with weight fragments in all thermo-regions, can be seen as the molecular conveyor-belt of the plant. The bonds between cellulose wall under construction and the lignin scaffoldings, also known as ligno-cellulosic structures, lead to higher onset temperatures in inner leaves. In the outer leaves, the cellulose structure is finished, with higher cellulose percents, and the peripheral lignin scaffoldings are partially disassembled and further converted to plants' needs or are found as fragments into the complex structure of lignin. The weight loss variations in the 9-proposed thermo-regions suggest an accelerated development of the molecular building blocks by foliar treatments with Se-474 VF. TGA also confirmed the FTIR and XRD results on pectin and cellulose Iα accumulation in the particular T4 and T5 thermo-regions presented in Table 2.”
Response 13. Thank you for your comment. We simplified the paragraph to be easier to understand (lines 591-600).
Point 14. Plants are not made from lignin and cellulose, certainly not herbaceous leafy plant as cabbage! What peripheral lignin scaffoldings exactly? It seems the authors are not familiar with plant structure, or cell wall structure, or understand there are many cell types and cell wall types….
Response 14. Thank you for your observation, but we do not fully agree with all the affirmations. We agree we should limit the discussion to the compositional aspect and we modified this. But cabbage contains cellulose and a small amount of lignin, and this was previously reported by other groups, which we added in the manuscript (lines 452-454). Additionally, we performed extractions of pectin, cellulose and lignin from cabbage, and partially characterized them as the revision time permitted. The FTIR and XRD analysis showed that the fractions extracted were as expected and a new experimental sub-section entitled "4.5. Extraction of pectin, cellulose and lignin from cabbage" was added (lines 989-1005). The lignin content in cabbage is enough to be visible by the techniques employed. We added at lines 442-444 that the lignin peaks have low intensities, suggesting small contribution (and in agreement with the extraction). Our intention was to spotlight better the significance of lignin for plant response to different stress – Moura, J. C. M. S., Bonine, C. A. V., de Oliveira Fernandes Viana, J., Dornelas, M. C., & Mazzafera, P. (2010). Abiotic and biotic stresses and changes in the lignin content and composition in plants. Journal of integrative plant biology, 52(4), 360-376. Liu, Q., Luo, L., & Zheng, L. (2018). Lignins: biosynthesis and biological functions in plants. International journal of molecular sciences, 19(2), 335. Cesarino, I. (2019). Structural features and regulation of lignin deposited upon biotic and abiotic stresses. Current opinion in biotechnology, 56, 209-214.
We agree that there are many cell and cell wall types – however, there is also an unity in diversity of the molecular logic of life. The lignin accumulation in plant cell wall is a response to stress also in plants from Brassicaceae family. Here we mention the following papers - Lahlali, R., Song, T., Chu, M., Yu, F., Kumar, S., Karunakaran, C., & Peng, G. (2017). Evaluating changes in cell-wall components associated with clubroot resistance using Fourier transform infrared spectroscopy and RT-PCR. International Journal of Molecular Sciences, 18(10), 2058 (the authors demonstrate that “Between susceptible (S) and resistance (R) samples, the most notable biochemical changes were related to an increased biosynthesis of lignin and phenolics”); Ahmad, J., Ali, A. A., Al-Huqail, A. A., & Qureshi, M. I. (2021). Triacontanol attenuates drought-induced oxidative stress in Brassica juncea L. by regulating lignification genes, calcium metabolism and the antioxidant system. Plant Physiology and Biochemistry, 166, 985-998; Dawood, M. F., & Azooz, M. M. (2019). Concentration-dependent effects of tungstate on germination, growth, lignification-related enzymes, antioxidants, and reactive oxygen species in broccoli (Brassica oleracea var. italica L.). Environmental Science and Pollution Research, 26(36), 36441-36457.
Point 15. Assigning special properties to molecules at different T regions is far fetched in my opinion. There’s no proof whatsoever for any influence on stress resistance at all.
Response 15. TGA analyses on commercial controls like pectin, microcrystalline cellulose and lignin have the role of supporting the main assignment of particular TGA regions. Of course, the decompositions can overlap or be shifted in the complex structure of plant cell wall, and the aim was not to assign the degradation steps to individual compounds, but to show an approximated thermogravimetric behavior and the changes induced in less stable and more stable compounds. The relevance of TGA is more related to the changes in the macromolecular composition as effect of Se-VF treatment and may not directly evidence a stress resistance.
Point 16. “All inner leaves have higher WL than outer leaves, which suggests that the biochemical processes that synthesize soluble fibers and proteins are more intense in the young leaves, and it can also be seen as an intermediary biosynthesis 501 step in the development of the plant, i.e., soluble fibers transformation into insoluble, structural fibers like cellulose and lignin”
- again, completely far fetched. No proof for that, and the rest of the data does not support your hypothesis. Please show a difference in protein levels.
Response 16. We changed to “soluble fibers and/or proteins”. The total protein amount has been determined by Kjeldahl method and presented at lines 341-344. The protein content increased significantly at D2 dose treatment. We changed “transformation” with “replacement” to avoid misunderstanding. We did not mean to say that the soluble fibers transform themselves as per se into cellulose and lignin, but that the different proportions change.
Point 17. “As proposed, if we see the biosynthesis of soluble fibers as an intermediary step in plant development, it means that our Se-BNF accelerates the plant metabolism through higher biosynthesis of volatile metabolomics, polyphenols, amino acids and peptides (T1-T2), faster conversion into thermolabile fibers such as proteins and oligosaccharides (T3) and further into more thermo stable soluble fibers, insoluble and structural fibers, fact evidenced by the lower content in soluble fibers in the fertilized samples in region T3, and confirmed in the next two thermo-regions T4 and T5 by higher content in hemicelluloses, pectins and cellulose in 515 treated samples.”
– the authors propose pectins (or as they call them – soluble fibers)- are intermediate step?????? It seems the authors never read a book on cell wall composition. Pectins are a constant part of any primary cell wall, while in cabbage those are the majority of cell walls.
Response 17. We tried to clarify the description for misunderstandings, since it is known that pectins are distinguishable structural biomolecules and a constant part of primary cell wall. But it is also known that pectins are diverse in composition and macromolecular structure, their common particularity being the galacturonic acid core. Low molecular weight pectins are hydrosoluble and may represent an intermediary step in plant cell wall development as primary cell wall, while high molecular weight pectins are less soluble and contribute to the middle lamela. Soluble fibers encompass soluble pectins, but not exclusively pectins. Therefore, the aim was to describe the intermediary step of pectins development from soluble, low molecular weight pectins to insoluble, high molecular weight pectins, in correlation with the development of primary cell wall, middle lamela and secondary cell wall. The plant cell wall undergoes numerous changes during plant development and as response to environmental conditions. We changed “biosynthesis” with prevalence (line 636) and added “partial replacement” (line 640) to avoid misunderstandings. It was never our intention to suggest that pectins are no longer part of the cell wall structure.
Point 18. Comments on the Quality of English Language: Some mistakes with plural and singular, and several bad phrasings.
Response 18. Hopefully, we managed to correct the mistakes and bad phrasing.
Reviewer 3 Report
The manuscript presents the biostimulant effect of the Vinase mixture with selenium on the productivity and structural composition of cabbage. The spectroscopic methodology used is novel and very useful to determine the structural changes of the leaves, and the modification of their main components. The results obtained contribute with basic knowledge about the physiology of the plant, and the beneficial effects of the biostimulant evaluated.
In the present work the ATR accessory was used, however, in the analysis of complex samples such as tissues or powders, the Diffuse Reflectance accessory is recommended, because it has a greater qualitative recovery of the information of functional groups, due to partially penetrates the sample. Based on the above, it is recommended to limit the scope of the discussion and conclusions to a qualitative and comparative level.
Quantitative infrared analyzes require a matrix with simultaneous variations of components for a chemometric analysis, therefore, in the present work, the scope of the results must be limited to only qualitative variation.
It would be convenient to have a spectrum of Glycinbetaine to compare with the spectra obtained in the leaf samples, in this way it could be determined if the increase of some signals, especially those registered in the range of 2920 cm-1 corresponds to the CH2 of the Glycinbetaine.
The discussion and conclusions about the stabilization of pectin in its demethylated form requires more spectroscopic evidence, since if the pectin methylation percentage decreases, the carbonyl signals of carboxylic acids would increase significantly, which does not happen. The absence of carboxylic acid signals is common when they are in the form of salts, therefore, the acidification of the sample to show the acids of the pectin can confirm their conclusions.
It is recommended to verify the numbering of Figures and tables throughout the entire manuscript.
Page 2, line 48. It is important to clarify that selenium does not have an antioxidant effect by itself. Selenium is part of an enzymatic system that reduces reactive chemical oxygen species.
Page 2, lines 52-53. What do they mean by "disproportionation reaction"?
Page 2 Line 69. What do they mean by acronyms"salts -BYV"?
Page 2, line 78. What do they mean by "macroergic"?
Page 2, Line 90. Change "microelement" to microelements
Page 4, Lines 129-130. Graph 5 and Table 3 do not seem to be related to their description in the text.
Page 6, Line 174. Figure 5 does not seem to be related to its description in the text.
Page 7, Line 232. Separate "(nano)biocomposites"
Page 9, Line 263. The number in the Figure is incorrect.
Page 9, Line 299. Must be Figure 4.
Page 9, Line 299. Table 4 does not exist in the text.
Page 12, Line 373. Separate "groupsThe"
Author Response
productivity and structural composition of cabbage. The spectroscopic methodology used is novel and very useful to determine the structural changes of the leaves, and the modification of their main components. The results obtained contribute with basic knowledge about the physiology of the plant, and the beneficial effects of the biostimulant evaluated.
Response 1. We thank you very much for your valuable description of our main aim of the study. We used your feedback to better define the aim and the hypothesis of this study.
Point 2. In the present work the ATR accessory was used, however, in the analysis of complex samples such as tissues or powders, the Diffuse Reflectance accessory is recommended, because it has a greater qualitative recovery of the information of functional groups, due to partially penetrates the sample. Based on the above, it is recommended to limit the scope of the discussion and conclusions to a qualitative and comparative level.
Response 2. Thank you for your recommendation. We limited our discussion and conclusion to the qualitative and comparative level.
Point 3. Quantitative infrared analyzes require a matrix with simultaneous variations of components for a chemometric analysis, therefore, in the present work, the scope of the results must be limited to only qualitative variation.
Response 3. Thank you for your recomendation. We modified.
Point 4. It would be convenient to have a spectrum of Glycinbetaine to compare with the spectra obtained in the leaf samples, in this way it could be determined if the increase of some signals, especially those registered in the range of 2920 cm-1 corresponds to the CH2 of the Glycinbetaine.
Response 4. The FTIR spectrum and XRD diffractogram of glycinebetaine-containing BYV were added in the Supllementary Material and discussed in the manuscript (lines 773-777).
Point 5. The discussion and conclusions about the stabilization of pectin in its demethylated form requires more spectroscopic evidence, since if the pectin methylation percentage decreases, the carbonyl signals of carboxylic acids would increase significantly, which does not happen. The absence of carboxylic acid signals is common when they are in the form of salts, therefore, the acidification of the sample to show the acids of the pectin can confirm their conclusions.
Reponse 5. Thank you very much for this useful observation. We are not sure we completely understood the request. We did some literature research in this respect as well. We found that in the case of pectin salts, the main peak of ionized carboxylic form is around 1600 cm-1 (we made a mistake in assigning it to COOH as well). The esterified form of hemicelluloses is in our case at 1738 cm-1 and this band should also include the COOH functional group. This would mean indeed that the decrease of the 1738 cm-1 peak accompanied by the increase of the 1600 cm-1 peak does not necessarily mean a deesterification, but it also does not exclude it. It also means that the methylation degree would be overestimated, therefore we reformulated the text both in the Results and in the Discussion, by using a more nuanced expression form and we mentioned that the peak area at 1738 cm-1 represents in fact esterified and protonated form of carboxyl (lines 504-507). Nevertheless, we think that the protonated form is in minority, based on the peak around 837 cm-1 (which should have some contribution from COOH as well), which is relatively small reported to other peaks and which increases significantly upon acidifiation. We mentioned this at lines 507-510 and 528-529.
We performed the acidification of the samples with 1 M H2SO4 and we presented and discussed these new FTIR spectra following your recommendation (lines 520-530). The acidification seems to decrease the COO- peak in D2, as expected, and probably increase the COOH peak (this is not really visible in all cases and it is difficult to determine quantitatively). The maximum of the right peak is now at 1638 cm-1 instead of 1610-1616 cm-1, which we believe indicates that the amide I has the highest contribution after acidification. This shift has been reported previously upon acidification.
But we do not understand how acidification can help distinguish between the presence and absence of demethylation. We believe other techniques are needed for this, but there was not enough time to perform them at revision and we do not know if the remained sample is enough.
Finally, we did not understand to what carbonyl signals that should increase does the Reviewer refer, we kindly ask to help us with more details.
Point 6. It is recommended to verify the numbering of Figures and tables throughout the entire manuscript.
Response 6. All the numbering of Figures, Tables and Sections were revised.
Point 7. Page 2, line 48. It is important to clarify that selenium does not have an antioxidant effect by itself. Selenium is part of an enzymatic system that reduces reactive chemical oxygen species.
Response 7. Thank you for your comment. In terrestrial plants there are no evidences up to now of enzymatic selenoproteins involved in reducing the reactive oxygen species – therefore we added ”by enhancing the activity of plant antioxidant systems.”
Point 9. Page 2, lines 52-53. What do they mean by "disproportionation reaction"?
Response 9. We added ” i.e., redox reaction wherein one compound (e.g., Se0) acts both as reductant and oxidant and generates two reaction products, one with a higher oxidation state (e.g., Se+4) and one in a lower oxidation state – e.g., Se-2.”
Point 10. Page 2 Line 69. What do they mean by acronyms"salts -BYV"?
Response 10. We rephrased it as "selenium salt and BYV".
Point 11. Page 2, line 78. What do they mean by "macroergic"?
Response 11. We added ”high-energy bonds, used for storing and transferring free energy in biological systems.”
Point 12. Page 2, Line 90. Change "microelement" to microelements
Page 4, Lines 129-130. Graph 5 and Table 3 do not seem to be related to their description in the text.
Page 6, Line 174. Figure 5 does not seem to be related to its description in the text.
Page 7, Line 232. Separate "(nano)biocomposites"
Page 9, Line 263. The number in the Figure is incorrect.
Page 9, Line 299. Must be Figure 4.
Page 9, Line 299. Table 4 does not exist in the text.
Page 12, Line 373. Separate "groupsThe"
Response 12. All the corrections were performed as suggested. Table 4 is present on page 20 and it is entitled "Comparison between variation in cellulose crystallinity as determined by FTIR and the XRD crystallinity". The number of Figures and Tables was revised throughout the manuscript.
Round 2
Reviewer 2 Report
The manuscript has improved significantly, I still find it very difficult to read- it's too long, and Discussion is extremely long and feels unfocused.
Author Response
Point 1. The manuscript has improved significantly, I still find it very difficult to read- it's too long, and Discussion is extremely long and feels unfocused.
Response 1. Thank you again for all the suggestions and recommendations that helped us to improve the manuscript. We revised the Discussion part again and moved part of the TGA and XRD that were mainly related to numerical aspects, from Discussion (lines 715-718, 747-768) to Results (lines 257-260, 345-356). We deleted some phrases that were repeating previous Information, e.g., lines 741-745. We also simplified and/or reformulated some phrases both in the Results and the Discussion part that we hope made the manuscript easier to follow. In the same time, we believe that it was important to describe and explain in detail the results, in order to sustain our claim that the methods can be used complementarily to more biologically-based methods. We also revised the section numbers, figures and tables for small mistakes.